



# The new TC-BC method and online instrument for the measurement of carbonaceous aerosols

Martin Rigler[1], Luka Drinovec[1,2], Gašper Lavrič[1], Athanasia Vlachou[3], André S. H. Prévôt[3], Jean Luc Jaffrezo[4], Iasonas Stavroulas[5], Jean Sciare[5], Judita Burger[6], Irena Kranjc[6], Janja Turšič[6], Anthony D. A. Hansen,[7] and Griša Močnik[1,2]

[1]Aerosol d.o.o., Ljubljana, Slovenia
[2]Jozef Stefan Institute, Ljubljana, Slovenia
[3]Paul Scherrer Institute, Villigen, Switzerland
[4]Univ. Grenoble Alpes, CNRS, IRD, G-INP, IGE, Grenoble, France
[5]Energy, Environment, Water Research Center, The Cyprus Institute, Nicosia, Cyprus
[6]Slovenian Environment Agency, Ljubljana, Slovenia
[7]Magee Scientific Corp, Berkeley, CA, USA

*Correspondence to*: M. Rigler (martin.rigler@aerosol.eu)

**Abstract.** We present the newly developed Total Carbon Analyzer (TCA08), and a new method for online speciation of carbonaceous aerosol with a high time resolution. The total carbon content is determined by flash heating of a sample collected on a quartz-fiber filter with a time base between 20 min and 24 h. The limit of detection is approximately 0.3 μgC, which corresponds to a concentration of 0.3 μgC/m$^3$ at a sample flow rate of 16.7 LPM and a 1-hour sampling time base. The concentration of particulate organic carbon (OC) is determined by subtracting black carbon concentration, concurrently measured optically by an Aethalometer®, from the total carbon concentration measured by the TCA08. The combination of TCA08 and Aethalometer (AE33) is an easy-to-deploy and low maintenance continuous measurement technique for the high time resolution determination of organic and elemental carbon (EC) in different particulate matter size fractions, which avoids pyrolytic correction and need for high purity compressed gases. The equivalence of this new online method to the standardized off-line thermo-optical OC-EC methods was evaluated during a winter field campaign at an urban background location in Ljubljana, Slovenia. The organic matter-to-organic carbon ratio obtained from the comparison with an Aerosol Chemical Speciation Monitor (ACSM) was OM/OC = 1.8, in the expected range.

## 1. Introduction

Carbonaceous aerosols frequently account for a large and often dominant fraction of fine particulate matter (PM2.5) mass in polluted atmospheres. They are extremely diverse (Gelencsér, 2004; Karanasiou et al., 2015) and they directly impact air quality, visibility, cloud formation and properties, the planetary radiation balance, and public health (Pöschl, 2005). The carbonaceous fractions can be described as black (BC) or elemental (EC) carbon, and organic matter (OM). OM is made up of many different molecular structures and includes not only particulate organic carbon, but also hydrogen, oxygen, nitrogen, and sulfur (Brown et al., 2013; Crenn et al., 2015). The amount of carbon that can be found in carbonaceous aerosols is called total carbon (TC), which is commonly categorized into fractions of organic carbon (OC) and elemental carbon (EC). OC can be directly emitted to the atmosphere in particulate form as primary organic matter by combustion and biogenic processes, or it can have a secondary origin from gas-to-particle conversion of (semi)volatile organic compounds in the atmosphere to aerosols after oxidation and condensation/nucleation (Hallquist et al., 2009). EC, on the other hand, is a mixture of graphite-like carbonaceous matter and is exclusively of primary origin and emitted by the incomplete combustion of carbonaceous fuels (Fuzzi et al., 2006; Karanasiou et al., 2015; Xu et al., 2015).





The first thermo-optical method for OC and EC determination was developed in 1982 by Huntzicker *et al*. (Huntzicker et al., 1982; Malissa et al., 1972). In thermo-optical methods, the carbonaceous aerosol deposited on the quartz filter is thermally desorbed according to a prescribed temperature protocol, first in an inert atmosphere (helium) and then in an oxidizing atmosphere (2% oxygen, 98% helium) (Cavalli et al., 2010). EC is thermally refractive and does not volatilize in an inert

atmosphere below ~700°C and can be combusted by oxygen at temperatures above 340°C (Karanasiou et al., 2015; Petzold et al., 2013; Schmid et al., 2001). Ideally, the OC fraction would desorb in the inert stage of the analysis, while EC would desorb and combust in the high temperature oxidizing stage of the analysis. Nevertheless, thermally unstable organic compounds pyrolyze (char) in the inert atmosphere to form pyrolytic carbon (PC), which combusts in the He+O$_2$ gas stream in a manner similar to the original EC (Cavalli et al., 2010; Karanasiou et al., 2015; Schmid et al., 2001). The PC that is formed during

analysis, if not properly accounted for, would be incorrectly reported as EC. To account for this, illumination by a laser beam is used to monitor the optical properties of the filter during the analysis by measuring reflectance or transmittance (Chow et al., 1993). Because PC absorbs light, light transmission and reflectance signals decrease during the inert stage of the analysis when the PC is created; and increase again in the oxidizing stage as the remaining carbonaceous material is burnt off the filter. The time when the reflectance or transmittance signal values meet the pre-pyrolysis value is called the OC-EC split point.

The three most commonly used thermal protocols are IMPROVE_A, NIOSH 5040 and EUSAAR2. The IMPROVE protocol using light reflectance for correction was designed to be applied to the Interagency Monitoring of Protected Visual Environments network in USA by Chow et al. (Chow et al., 1993). The NIOSH protocol using light transmittance was developed for the analysis of the carbonaceous fraction of particulate diesel exhaust based on the U.S. National Institute of

Occupational Safety and Health method 5040. In 2010, the thermo-optical analysis protocol EUSAAR2 was developed for European regional background sites. In order to improve the accuracy of the OC-EC split of this protocol, lower temperature steps in the inert stage of the analysis and longer residence times are used to achieve reduction of PC and more complete evolution of OC (Cavalli et al., 2010). This protocol has recently became part of the European standard for the determination of OC-EC in PM2.5 samples (EN 16909:2017, 2017).

The charring of organic material during thermal analysis is an important uncertainty of the thermo-optical methods. tTe amount of OC converted into PC during the analysis depends on many factors, including the amount and type of organic compounds, the sources of air pollution, temperature steps in the analysis, the residence time at each temperature step, and the presence of certain inorganic constituents (Yu et al., 2002). When correcting for PC, thermal-optical methods make two important

assumptions:

    (1) PC created by charring during the helium stage of the analysis is more easily oxidized and will evolve before the original EC

    (2) The specific light attenuation cross section of PC ($\sigma_{PC}$) is similar to that of the original EC on the filter ($\sigma_{EC}$).

However, PC and original EC combust concurrently in the oxidizing stage of the analysis. Moreover, PC can evolve even

prematurely in the inert atmosphere depending on the thermal protocol used for the analysis, especially in the presence of oxygen donor substances in the sample (Sciare et al., 2003). Additionally, PC and EC have been shown to have significantly different values of $\sigma$ (Bhagawan et al., 2015a; Cavalli et al., 2010; Chen et al., 2014; Karanasiou et al., 2015; Subramanian et al., 2004). The $\sigma_{PC}$ is mostly affected by the composition of its organic precursors, aerosol type and duration of sampling. For this reason, the magnitude of the uncertainty of the OC-EC split point varies from one aerosol sample to another. Overall, the

uncertainty derived from an incorrect determination of the OC-EC split is a function of the following parameters (Karanasiou et al., 2015):

    -   Aerosol type: the amount of PC converted from OC in the sample and its properties

    -   Sample oven soiling (i.e., presence of catalytic residues)





- Interference from other aerosol components: Carbonate carbon, Metal Oxides, Inorganic salts, Brown carbon
- Thermal protocol used for analysis.

Because OC is the larger and often the dominant fraction of TC, the uncertainty from an incorrect OC-EC split point has a
greater effect on the EC value. However, TC is a measurement of all evolved carbon, irrespective of the possible conversion
of the fractions or the sample properties. Hence the TC determination is not influenced by the amount of PC formed during
analysis or the thermal protocol used,  and is therefore independent of the parameters mentioned above.

Thermal and optical methods refer to different properties of carbonaceous aerosol and specific attention needs to be paid to
use appropriate terminology when inter-comparing carbonaceous analysis techniques using different measurement methods
(Petzold et al., 2013). Measurements of optical attenuation or absorption are converted to mass concentration of black carbon
(BC) using an externally determined mass attenuation/absorption cross-section – the resulting quantity is called equivalent
black carbon (eBC, Petzold et al., 2013). The thermo-optical and optical measurements share more than the optical pyrolysis
determination during the inert phase of the heating in a thermal-optical analyzer. The definition of eBC is tied to the thermal
determination of the sample carbon content – the sample optical attenuation was compared to its thermally determined carbon
content, both analyses performed after Soxhlet extraction (to remove non-soluble carbon), obtaining the BC mass attenuation
cross-section independent of a specific thermal protocol (Gundel et al., 1984).

It was shown that the soluble carbon fraction did not absorb significantly, as the attenuation for the extracted samples decreased
by no more than 7% compared to the non-extracted ones. While the insoluble fraction is not identical to the thermally refractive
one, the relationship between the optically determined BC and the thermo-optically determined EC can be determined by
analyzing samples obtained at the same site during the same period. Differences in thermal protocols, giving (systematically)
different EC values (Bae et al., 2009; Karanasiou et al., 2015), will result in different EC-to-BC regression slopes. At the same
time, differences in the sample composition (and the sources of the aerosols) will influence the OC-EC split point, resulting in
evolution of the less refractive part of EC in the inert phase and the more refractive part of OC in the oxidizing phase
(Karanasiou et al., 2015). Sample composition and sources also impact the sample optical properties, especially at shorter
wavelengths (Sandradewi et al., 2008; Zotter et al., 2017). All of these factors affect the relationship between EC and BC.

Carbonaceous aerosols are the major, dominant component of the mass of suspended particles in polluted atmospheres.
Accurate, continuous and high time resolved data are needed in order to assess the severity of the problem and to identify and
investigate the main sources which require attention; and to quantitate the improvements following the application of controls
and regulations. The new TC-BC method presented in this study is an easy-to-deploy and low maintenance continuous
measurement technique for the high time resolution determination of organic and elemental carbon in different PM fractions
(PM10, PM2.5 and PM1). It can be used for routine air quality monitoring applications, field work and laboratory research.
For example, high-time resolution data from the TC-BC method in combination with different size selective inlets can be used
for quality control in aerosol mass spectrometry through comparison of differently derived oxygen to carbon (O-C) and organic
aerosol to organic carbon (OA-OC) ratios (Pieber et al., 2016). In this study, the new online TC-BC method was tested during
a field campaign from 7 February to 10 March 2017 at an urban background air quality monitoring station of the Slovenian
Environmental Agency (ARSO).  High time resolved data of TC and BC were compared to EUSAAR2 OC-EC analysis of
$PM_{2.5}$ filter samples that were collected in parallel with a high volume sampler; and to organic aerosol mass measured by
ACSM with a $PM_1$ aerodynamic lens.  The equivalence of the new online TC-BC method to the standardized off-line thermo-
optical OC-EC methods is evaluated through analysis of regression models of the various compared methods.





## 2. Method and instrument description

### 2.1 TC-BC method for online high time resolved OC-EC measurements

In this study we present the newly developed TC-BC method, which combines an optical method for measuring mass equivalent black carbon (eBC) by the AE33 Aethalometer (Drinovec et al., 2015; Hansen et al., 1984), and a thermal method

for total carbon (TC) determination by a new instrument, the Total Carbon Analyzer TCA08, developed and commercialized by Aerosol d.o.o. (Ljubljana, Slovenia). The TC-BC method determines equivalent organic carbon (eOC) fraction of carbonaceous aerosols defined as:

$$eOC = TC - eEC , \quad (2)$$

where

$$eEC = b \cdot eBC \quad (3)$$

is equivalent to elemental carbon (EC) and the determined proportionality parameter $b$ is region/site specific but also depends to a large extent on the thermal protocol used to determine the EC fraction with a conventional OC-EC method. We call this determined parameter 'equivalent elemental carbon' (eEC) since the measurement method is an optical one, and its result is converted to an equivalent concentration of elemental carbon, following the terminology logic of Petzold et al. (2013).

### 2.2 The TCA08 Total Carbon Analyzer

The TCA08 Total Carbon Analyzer instrument uses a thermal method for total carbon (TC) determination. The instrument contains two parallel flow channels with two analytical chambers, which alternate between sample collection and thermal analysis. While one channel is collecting its sample for the next time-base period, the other channel is analyzing the sample

collected during the previous period. This sequential feature offers the great advantage of a continuous measurement of TC. Fig. 1 (a) shows the TCA08 flow diagram, controlled by a system of valves which alternate the two channels to the common elements of pump, $CO_2$ analyzer, etc. The instrument collects the sample of atmospheric aerosols on a central spot area of 4.9 cm² of a 47-mm diameter quartz fiber filter enclosed in a small stainless-steel chamber (Fig. 1 (b)), at a controlled sampling flow rate of 16.7 LPM, i.e. 1 m³ per hour, provided by a closed-loop-stabilized internal pump. The sampling time may be pre-

set from 20 minutes to 24 hours. A 1-hour time-base was used in the studies reported here.

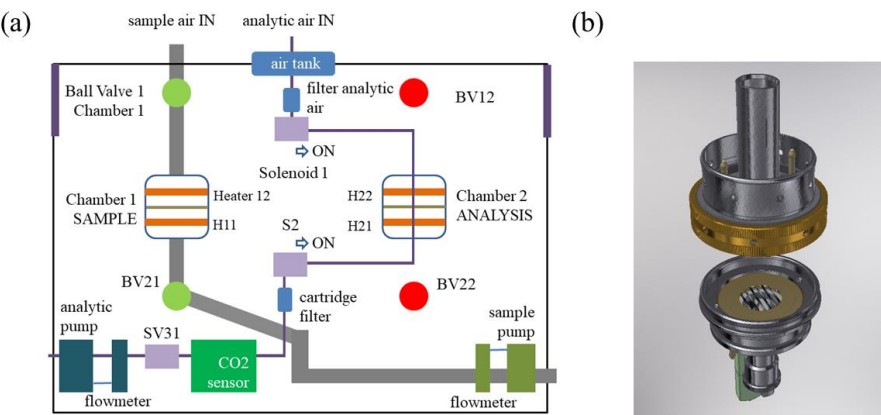

**Figure 1: (a) The TCA08 flow diagram.** While chamber 1 collects a new atmospheric sample on the quartz-fiber filter, chamber 2 performs a thermal analysis of the previously collected sample. The system of ball valves (BV11, BV21, BV12, BV22) and solenoids (S1 and S2) change the airflows after the sample time-base. **(b)** The analytical chamber of the TCA08 Total Carbon Analyzer is

made of stainless steel. It supports the quartz-fiber aerosol collection filter between two closely-spaced heating elements, one above and one below.





At the end of the collection period, the sample flow is switched from one channel to the other. A different configuration of valves provides a small analytical flow of 0.5 LPM of ambient air through the quartz-fiber filter and then to the $CO_2$ detector. Before entering the chamber, the analytic air passes through a 10-liter buffer volume for ambient $CO_2$ fluctuation averaging and a capsule filter filled with activated carbon and pleated glass fiber filter, which removes organic gases and particles from

the stream. High-power electrical elements above and below the quartz filter heat the sample almost instantaneously to 940°C, efficiently combusting carbonaceous compounds into $CO_2$. Since the amount of $CO_2$ produced is large compared to the internal volume of the system, this creates a pulse of $CO_2$ in the analytical air stream of short duration but well-defined amplitude over the baseline.

This has the very great advantage that filtered ambient air may be used as the analytical carrier gas, after temporal stabilization in the internal buffer volume to remove any rapid ambient fluctuations. This feature facilitates the field deployment of the TCA08 instrument, as it does not require compressed (carrier) gas for the analysis. The carrier gas concentration of $CO_2$ is measured before and after the combustion step and fit using a polynomial function to create the baseline. The increase in $CO_2$ concentration above baseline is measured and integrated to give the Total Carbon content of the sample ($m_{TC}$):

$$m_{TC} = C_{carb} \left\{ \int_{t_1}^{t_2} f_A(t) \left[ CO_2^{signal}(t) - CO_2^{ambient}(t) \right] dt - \int_{t_3}^{t_4} f_A(t) \left[ CO_2^{blank}(t) - CO_2^{ambient}(t) \right] dt \right\}, \qquad (4)$$

where $C_{carb}$ is a carbon calibration constant determined by a calibration with punches of ambient filters with known TC content; $t_2 - t_1$ is the combustion duration of heating 1; $f_A(t)$ is the analytical air flowrate during combustion; and $\left[ CO_2^{signal}(t) - CO_2^{ambient}(t) \right]$ is the $CO_2$ signal measured by the NDIR detector, relative to the fitted baseline level of $CO_2$ in the ambient air stream. The second heating ($t_4 - t_3$) is performed after the first heating when the chamber is cooled down to

room temperature again. Term $\left[ CO_2^{blank}(t) - CO_2^{ambient}(t) \right]$ is the CO2 blank filter measurement relative to the fitted baseline level of $CO_2$, as a result of NDIR detector artefact due to rapid change of the air temperature in the chamber. The duration of analysis is 17 min and includes two identical heating and cooling cycles with measurement of background $CO_2$ before and after heating. An example of such subtraction of two integrals in Eq. 4 is shown in Figure 2.

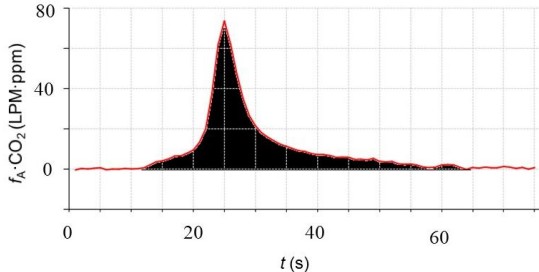

**Figure 2: Example output from the CO2 detector in the TCA08 Total Carbon Analyzer, showing the combustion-derived pulse of CO2 superimposed on the ambient-air baseline.**

The $CO_2$ sensor used in TCA08 is the LI-840A CO2/H2O Analyser (LICOR, Inc., 2016). It is an absolute, non-dispersive infrared gas analyser based upon a single path, dual wavelength and thermostatically controlled infrared detection system. Concentration measurements of $CO_2$ and $H_2O$ are based on the difference ratio in IR absorption between sample and reference signal. The $CO_2$ sample uses an optical filter centred at wavelength of 4.26 μm (reference at 3.95 μm), while for $H_2O$ at 2.595 μm (reference at 2.35 μm). The concentration measurement of $CO_2$ is pressure compensated and corrected for spectral cross-

sensitivity of water molecules with an uncertainty less than 1 ppm (at 370 ppm and 1 second signal filtering).





**2.3 Positive and negative sampling artifacts in the TCA08 Total Carbon Analyzer**

The measurement of carbonaceous aerosols using quartz-fiber filters is challenging because of the possibility of positive and negative sampling artifacts (Cheng et al., 2009; Kirchstetter et al., 2001; Subramanian et al., 2004; Watson et al., 2008). The adsorption of organic vapors (Volatile Organic Compounds, VOCs) onto quartz-fiber filters during aerosol sampling causes OC concentrations to be over-reported, while volatilization of the collected aerosols from the filter results in the loss of OC. These sampling artifacts have been estimated to range between +50% for adsorption (Arhami et al., 2006a; Kirchstetter et al., 2001) to -80% for volatilization (Modey, 2001). In the European standard (EN 12341:2014, 2014) this phenomenon is acknowledged but not considered in the uncertainty budget, as its magnitude cannot be quantified precisely. However, different studies of positive and negative sampling artifacts have shown that the magnitude depends on the sampling face velocity, sampling duration, filter substrate, pre-firing of filters, ambient temperature, and location with its characteristic aerosol type (Karanasiou et al., 2015; Mader, 2003; Subramanian et al., 2004; Turpin et al., 2000). For comparison purposes, table 1 shows a comparison of sample flow, sample face velocity, sample time-base and filter media for the two different filter based instruments used in this study; Digitel Sample DHA-80 (DIGITEL Elektronik, 2012) and the TCA08. Different studies have noted that adsorption tends to be the dominant artifact at low-volume ambient sampling and shorter sample time-bases. Consequently, we expect that volatilization effects will be small for the conditions used in the TCA08 instrument (McDow and Huntzicker, 1990; Subramanian et al., 2004; Turpin et al., 2000)

| Instrument | Exposed filter diameter $d$ [mm] | Flow [LPM] | Face Velocity [cm/s] | Sample timebase | Filter material |
|---|---|---|---|---|---|
| Digitel Sampler DHA-80 | 143 | 500 | 51.9 | 24 h | Quartz fiber |
| TCA08 | 25 | 16.7 | 56.7 | 20 min-24 h, this study 1 h | Quartz fiber |

**Table 1: Filter collection area diameter, sample flow rate, face velocity, sample time-base and filter material for the filter-based instruments used for the OC-EC concentration measurements**

Different approaches have been used to minimize the adsorption artifact and to quantify its magnitude: such as the "two filters" approach (quartz behind quartz, QBQ; quartz behind Teflon, QBT); the "slicing filters" approach; regression intercept approach; and the use of denuders (Eatough et al., 1999; Watson et al., 2008). For routine measurements in monitoring networks, a VOC denuder appears to be the most practical and realistic approach (Cavalli et al., 2016; Watson et al., 2009). Such denuders trap gaseous carbonaceous species, which would otherwise be adsorbed by quartz fiber filters and measured as a positive sampling artifact. The denuder adsorbs organic gases by diffusion to its wall surfaces, while the aerosols remain suspended in the sample stream and are unaffected. The TCA08 instrument uses a honeycomb charcoal denuder to remove gas-phase OC with high efficiency at the sampling flow rate of 16.7 LPM. Residence time for one denuder monolith in the TCA08 is 175 ms. Honeycomb denuders have a high density of channels and offer a large active surface area in a compact size (Mader et al., 2001). Additionally, solid charcoal material does not deteriorate under the influence of humidity, which is an advantage compared to denuders fabricated with carbon impregnated strips (Cavalli et al., 2016).

Depending on location and the concentration of organic gases, some VOCs can still penetrate through the denuder and be adsorbed by the quartz-fiber filter matrix (denuder breakthrough, Arhami et al., 2006b; Zhang et al., 2013). Denuder breakthrough occurs when the time for trapping VOCs is longer than the residence time. During the sampling the actual





capacity of the denuder slowly decreases, as the denuder surfaces become occupied by adsorbed VOC, leading to increased times to trap all VOC. Longer residence times are needed in such occasions (2 or more denuder monoliths). To account for this artefact, the TCA08 instrument incorporates a test procedure which can be used to determine the on-site efficiency of the VOC denuder and denuder breakthrough value on site. This (QBQ) approach integrates an in-line filter in the sample inlet
stream to remove filterable aerosols. The denuder is then installed in the flow stream passing to Channel 1, while Channel 2 receives the un-denuded stream (Fig. 3).

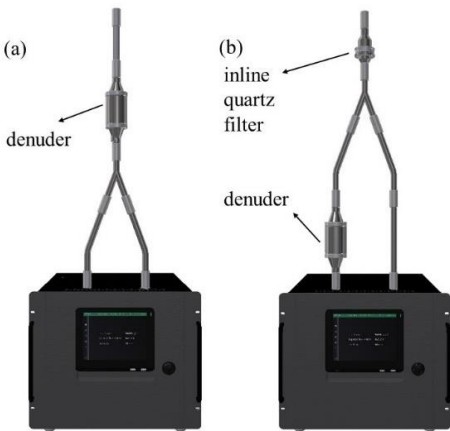

**Figure 3: TCA08 setup when (a) sampling and (b) performing denuder efficiency test. Note that the tubing length is identical in both setups. This permits the test to be performed at a permanent installation without disturbing the inlet plumbing.**

The denuder efficiency $E_D$ is determined by comparing the TC results in chamber 1 and chamber 2 as

$$E_D = \left[\frac{1}{n}\sum_n \frac{TC_{F,n} - TC_{F+D,n}}{TC_{F,n}}\right],\tag{5}$$

where $TC_{F+D,n}$ is $n$-th Total Carbon content measured in chamber 1, where air sample stream goes through filter above divider and denuder and $TC_{F,n}$ is $n$-th Total Carbon content measured in chamber 2, where air sample stream goes only through filter above divider. Constant gaseous OC concentration approximation through $n$ measurements is used for calculation. $TC_{F+D,n}$ also represents denuder breakthrough value.

We developed these routines during the instrument design and performed the measurements as part of the field campaign (see Section 3). After five weeks of continuous operation with consistent TC data (see below), the measured denuder efficiency was 74%. We recommend that the denuder should be replaced or regenerated when its efficiency drops below 70% (Ania et al., 2005; Bhagawan et al., 2015b; Gao et al., 2014). The Standard Operating Procedure for routine use of the TCA08 instrument recommends replacement or regeneration of the denuder honeycomb element once per month. Further, in
environments with high VOC concentrations, two denuder honeycombs in series are recommended (Gregorič et al., 2019).

**2.4 Field testing measurement campaign**

The TCA08 instrument was evaluated during a field measurement campaign at an urban background site in Ljubljana, Slovenia. Ljubljana is a city of ~350,000 inhabitants located at the southern edge of a geographic basin. In wintertime, it is





characterized by poor ventilation and frequent temperature inversions. Air quality in Ljubljana is influenced mostly by traffic and also by the combustion of biomass for household heating, both within the city and in surrounding areas (Ogrin et al., 2016). The measurement campaign was conducted between 7 February and 10 March 2017 at the urban background air quality monitoring station of the Slovenian Environmental Agency (ARSO) at 46.0654°N, 14.5120°E, elevation 299 m. This sampling

site and period of the year were selected to test the performance of the instrument in a complex environment characterized by various sources of carbonaceous aerosols (traffic, domestic heating, secondary organic) exhibiting strong temporal variability and a wide range of properties (OM/OC, OC-EC, volatility, etc). During the Ljubljana campaign, the daily average measured TC concentrations ranged from 3 to 26 µg/m³. This provided a wide dynamic range for the inter-comparison of methods and analyses.

The TCA08 was operated on a 1-hour time-base, sampling PM$_{2.5}$ fraction at 16.7 LPM; co-located with a Model AE33 Aethalometer measuring Black Carbon aerosols in PM$_{2.5}$ on a 1-minute time-base at 5 LPM. At the same location, 24-hour PM$_{2.5}$ filter samples were collected in parallel with a Digitel high volume sampler for OC-EC offline analysis at two different laboratories; the Slovenian Environmental Agency (ARSO, Ljubljana, Slovenia), and IGE (Grenoble, France) both using the

Sunset offline OC-EC analyzer with the EUSAAR_2 thermal protocol. Additionally, non-refractory organic matter (OM) measurements were also performed during the campaign with an ACSM (Aerodyne, Billerica, MA; Ng et al., 2011) on a 29-30 min time-base to derive high-time resolution measurements of the OM-to-OC ratio. The ACSM, equipped with a PM$_1$ aerodynamic lens, was sampling through a PM$_1$ sharp cut cyclone (SCC 1.197, BGI Inc.) at a flow rate of 3 LPM yielding a particle cut off diameter of roughly 3 µm. Furthermore, the sample was driven through a Nafion dryer, upstream the instrument

inlet, keeping the sample relative humidity below 40% throughout the campaign. The chemical composition dependent collection efficiency of the instrument was determined according to Middlebrook et al., 2012. Due to variability in the ACSM time-base, we gathered the data into 3h averages. All of the instruments were checked regularly and operated without interruption throughout the campaign. No data were selectively removed from the results presented in the following.

### 3   Results and discussions

Table 2 reports comparison results between offline filter measurements and 24 h average values of high time resolution measurements of TC, BC, TC-$b$BC and OM; and between high time resolution measurements (1 h) of TC-$b$BC and OM. Linear orthogonal regression results are shown with $s$ as the slope for the model without an intercept, and with $s_1$ as the slope and $i$ as the intercept for the model with an intercept (EN 16450:2017, 2017). $R_{xy}^2$ is the square of the Pearson correlation coefficient. 31 samples were collected for the offline comparison.


|  |  |  | Orthogonal regression results |  |  |  |  |  |
|  |  |  | $y = s \cdot x$ |  | $y = s_1 \cdot x + i$ |  |  | $b = 1/s$ |
| $x$ | $y$ | N | $R_{xy}^2$ | $s$ | $R_{xy}^2$ | $s_1$ | $i$ $[\mu g/m^3]$ |  |
| TC$_{ARSO}$ | TC$_{IGE}$ | 31 | 0.99 | 1.03 ± 0.01 | 0.99 | 1.10 ± 0.01 | -0.79 ± 0.14 |  |
| OC$_{ARSO}$ | OC$_{IGE}$ | 31 | 0.99 | 1.01 ± 0.01 | 0.99 | 1.09 ± 0.01 | -0.81 ± 0.12 |  |
| EC$_{ARSO}$ | EC$_{IGE}$ | 31 | 0.91 | 1.09 ± 0.03 | 0.94 | 0.99 ± 0.05 | -0.19 ± 0.07 |  |
| TC (see Eq.7) | TC$_{TCA08}$ | 31 | 0.98 | 1.00 ± 0.02 | 0.99 | 0.92 ± 0.02 | 0.99 ± 0.15 |  |
| EC (see Eq.7) | BC$_{AE33}$ | 31 | 0.87 | 2.27 ± 0.09 | 0.88 | 2.45 ± 0.15 | -0.36 ± 0.25 | **0.44 ± 0.02** |
| OC (see Eq.7) | OC$_{TC-BC}$ | 31 | 0.94 | 0.99 ± 0.02 | 0.98 | 0.86 ± 0.02 | 1.33 ± 0.18 |  |
| OC | OM$_{ACSM}$ | 31 | 0.97 | 1.79 ± 0.03 | 0.97 | 1.79 ± 0.05 | 0.07± 0.44 |  |
| OC$_{TC-BC}$ | OM$_{ACSM}$ | 300 | 0.96 | 1.82 ± 0.01 | 0.97 | 2.05 ± 0.02 | -2.45 ± 0.20 |  |





**Table 2: Summarized comparison results between off-line filter measurements and 24 h average values of high-time resolution measurements of TC, BC, $OC_{TC-BC}$ and OM; and between high time resolution measurements (3h) of $OC_{TC-BC}$ and $OM_{ACSM}$ measurements**

A more in-depth analysis of these different correlations is provided in the following.

**3.1 Inter-laboratory comparison of off-line carbon analyses of 24-hour filter samples**

Figure 4 shows the comparisons of the off-line measurements performed by the ARSO and IGE laboratories for TC (a), OC (b), and EC (c); the OC-EC split point was derived from the thermogram using the EUSAAR_2 thermal protocol.

(a)

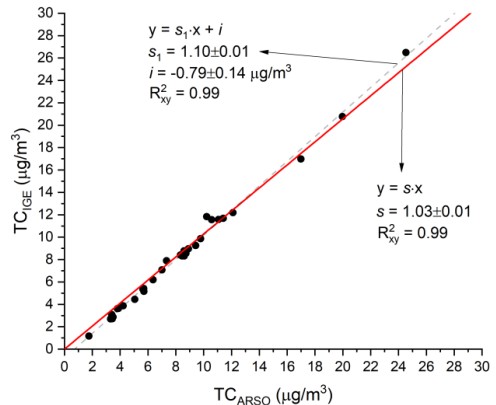

(b)

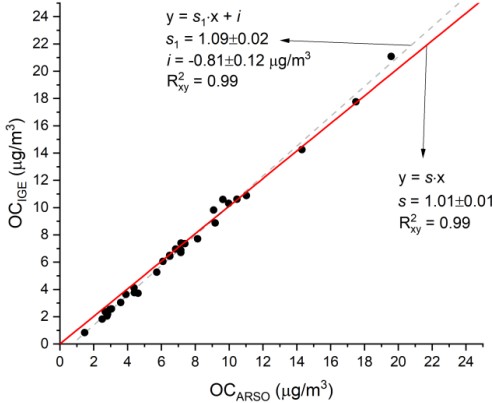

15    (c)



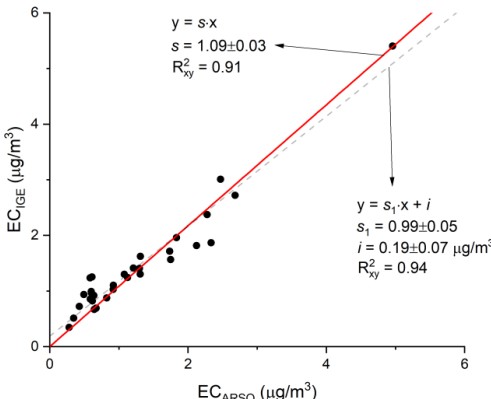

**Figure 4: Comparisons of offline measurements of (a) TC, (b) OC and (c) EC from the ARSO and IGE laboratory analyses. OC and EC were measured using the EUSAAR_2 thermal protocol. Linear orthogonal regression results are shown with $s$ as the slope (red line) for the model without an intercept and with $s_1$ as slope and $i$ as intercept (dashed gray line) for the model with an intercept. $R_{xy}^2$ is the square of the Pearson correlation coefficient. 31 samples were collected for analysis during the campaign.**

These results show that the off-line analyses of filter samples collected during the field campaign were consistent between the two external laboratories, both for the Total Carbon content of the samples, as well as for the partitioning into EC and OC components. The uncertainty $u_{RM}$ between the reference methods for TC

$$u_{RM}^2 = \frac{1}{2n} \sum_{i=1}^{n} \left( TC_{i,ARSO} - TC_{i,IGE} \right)^2 \tag{6}$$

is 0.43 μg/m³ which is well below the limit of 2.00 μg/m³ requested for reference methods (EN 16450:2017, 2017). However, the difference in slope for OC and consequently for TC is around 10%, with a negative intercept value of around -0.80 μg/m³ for OC and TC (using linear orthogonal regression model with intercept) which can indicate differences in instrument calibration, suboptimal performance of one of the instruments (featuring artefacts) or inadequate filter sample handling. The filter samples were first measured in ARSO laboratory, and then shipped to IGE laboratory. Sampling, transport and storage of the filters were done according to the 16909:2017 standard (EN 16909:2017, 2017).

These uncertainties and the regression slope are consistent with the results of the inter-laboratory comparisons conducted in the ACTRIS (Aerosol, Clouds and Trace Gases Research Infrastructure) framework, where TC repeatability (intra-laboratory measurement comparison) and reproducibility (inter-laboratory measurement comparison) were reported to be in the range of 2% – 6% and 3% – 13%, respectively (ACTRIS, 2016, 2017, 2018). For EC/TC, the ACTRIS exercises gave much larger reproducibility percentages, so, while there seems to be here a systematic (about 10%) difference between the two laboratory analyses, the difference is within the range expected for the OC-EC determination. The OC-EC determination is quality controlled in the comparison exercise in which the Slovenian laboratory was participating. The 10% difference in TC is larger than the reproducibility and repeatability of urban background samples analyzed in this exercise , and the difference is smaller for EC (ACTRIS, 2016) . This leads us to conclude that while the differences between the laboratories can be large, the 10% difference between two laboratories using the same thermal protocol and sample protocols according to the applicable standard (EN 16909:2017, 2017) is not unusual (Panteliadis et al., 2015).

To reduce the uncertainty of OC-EC data in further analysis, an average of TC, OC and EC measurements on filters from both laboratories is used and reported in Table 2. Consequently, daily filter values of $TC_i$, $OC_i$ and $EC_i$ are defined as



$$TC_i = \left(TC_{i,\text{ARSO}} + TC_{i,\text{IGE}}\right)/2,$$
$$OC_i = \left(OC_{i,\text{ARSO}} + OC_{i,\text{IGE}}\right)/2,$$
$$EC_i = \left(EC_{i,ARSO} + EC_{i,\text{IGE}}\right)/2, \tag{7}$$

where $1 \leq i \leq 31$ represent each 24 h filter during the measurement campaign.

### 3.2 Comparison of TC on-line measurements with off-line filter analyses

Figure 5 shows a time series comparison of the 1-hour and 24-hour average TCA08 data, together with the offline analyses results for TC analysis of filter samples defined by Eq. 7. Gaps in the TCA08 measurement data are due to regular maintenance and quality control procedures (quartz filter change procedure, denuder efficiency test, *etc*)

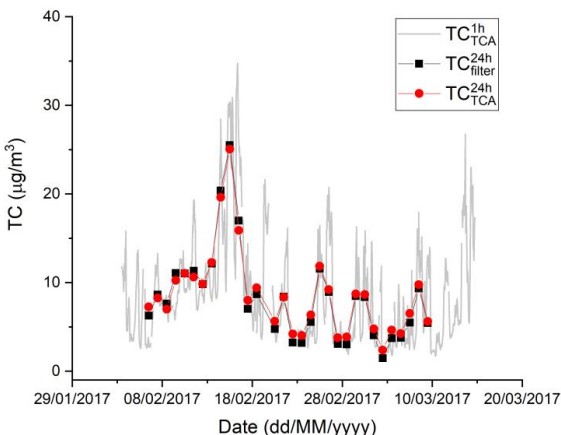

**Figure 5: Time series comparison of off-line results for TC derived from offline filter analyses; to 1-hour and 24-hour averaged TC data from the on-line TCA08 measurements.**

These results show that on-line operation of the new TCA08 instrument with its simplified analysis method agrees very well

20 with TC data measured by off-line **thermo-optical** analyses of filters. Figure 6 shows the comparison of these two datasets.





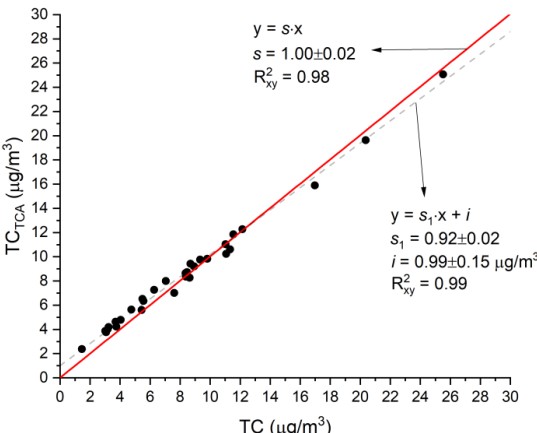

**Figure 6: Comparison of offline measurements of TC (laboratory filter analyses), to the 24-hour average of 1-hour online measurements of TC from the TCA08. Linear orthogonal regression results are shown with *s* as the slope (red line) for the model**

**without an intercept and with $s_1$ as slope and *i* as intercept (dashed gray line) for the model with an intercept. $R_{xy}^2$ is the square of the Pearson correlation coefficient. 31 samples were collected for analysis during the campaign.**

The correlation plot of 24 h average TC results from the TCA08 versus the TC analyses of offline filters show high Pearson correlation coefficients ($R_{xy}^2$ above 0.98 for both regression models). Linear orthogonal regression model without intercept

shows slope *s* equal to $1.00 \pm 0.02$, while model with the intercept shows slope $s_1 = 0.92 \pm 0.02$ and intercept of $0.99 \pm 0.15$ $\mu g/m^3$.

The fact that these slopes are close to unity for both regression models, shows that the of TCA instrument using no catalyst and filtered ambient air as the carrier gas during analysis, has as high a combustion efficiency as the conventional offline OC-

EC analyser. The intercept of $0.99 \pm 0.15$ $\mu g/m^3$ may indicate a positive sampling artefact as described in Chapter 2.3. The positive sampling artifact attributed to VOC adsorption is more pronounced for the TCA method compared to offline filter analysis due to the difference in the sampling time, since both methods use similar face velocity (Table 1). VOC adsorption is most pronounced at 1 h sampling time and saturates in a few hours (Gregorič et al., 2019); with a 24 h sampling time, the VOC contribution is small. Over a period of 24 hours, VOCs adsorbed onto the filtercan during cooler parts of the day may be

desorbed during warmer parts of the day, reducing their contribution to the OC result. The contribution of positive and negative artefacts for the 24 h filters is hard to estimate, while for short sample time base the positive artefact prevails and can be described with a saturation curve. Therefore, the measured offset can be accounted for by denuder breakthrough, which was measured and confirmed by the denuder efficiency test. The delta analysis between TC analysis done on 24h offline filters and online TC with 1 h time resolutions confirms this phenomenon, especially for the days with lower total carbon concentrations

(lower than 5 $\mu g/m^3$), where the relative difference between both methods can reach 25-50 % (Figure 9). To achieve a lower offset in comparison to OC-EC measurements based on 24 h filters for the sampling sites with lower concentrations of TC, two denuder monoliths or a longer sampling time base should be used.

.




### 3.3 TCA08 method uncertainty

The uncertainty of TC data from conventional OC-EC analyzers is determined by the uncertainty of the volume of injected gaseous standard at the end of each analysis; the uncertainty of the external calibration standard; and the uncertainty of the $CO_2$ and flow measurements during analysis (EN 16909:2017, 2017). The uncertainty $u_{TCA}$ associated with the TC data from

the TCA08 includes individual uncertainty sources of the carbon calibration factor $C_{carb}$; the uncertainty of the analytic flow measurement; and the uncertainty of the signal and blank $CO_2$ peak measurement (Eq. 4). To calculate the measurement uncertainty of data from the TCA08, the $CO_2$ signal measured by the NDIR detector is approximated with a box function, with its integral value the same as of the measured $CO_2$ signal function (Fig. 3). The height of the $CO_2$ box function is a linear function of TC mass collected on the filter. The relative uncertainties of $C_{carb}$ and analytic flow are determined to be 5% and

2%, respectively, while the absolute uncertainty of $CO_2$ measurement is approximately 1 ppm. The $u_{TC}$ for a representative range of concentrations of TC in air, using a 1h timebase and sampling at 16.7 LPM, is estimated to be

$$u_{TCA}[\text{LoD}=0.3\ \mu g/m^3] = 41\ \%,$$
$$u_{TCA}[\text{TC} = 2.5\ \mu g/m^3] = 6\ \%,$$
$$u_{TCA}[\text{TC} = 10\ \mu g/m^3] = 3\ \%, \tag{8}$$

where LoD is the limit of detection of the TCA08 at a sample flowrate of 16.7 LPM and sample timebase of 1h.

### 3.4 Comparison of on-line BC measurements with off-line EC filter analyses

Figure 7 shows the regression of the off-line thermo-optical analysis of samples for EC (from the ARSO and IGE laboratories, using the EUSAAR_2 protocol) with the 24-hour averaged BC (Aethalometer data) obtained during the field campaign period. The Pearson correlation coefficients of 0.87 and 0.88 are very similar for each of the regression models (with/without

intercept). The linear relationship between EC and BC is described by slope $s$ when using orthogonal regression model without intercept. The proportionality parameter $b$ (Eq. 3) is determined as

$$b = \frac{1}{s} = 0.44 \pm 0.02 \ . \tag{9}$$

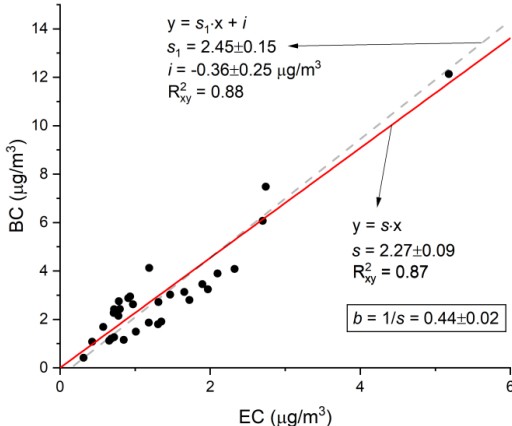

**Figure 7: Comparisons of offline measurements of EC (laboratory filter analysis) using the EUSAAR_2 thermal protocol, to the 24-hour average of online measurements of BC data taken by the AE33 Aethalometer. Linear orthogonal regression results are shown with s as the slope (red line) for the model without an intercept and with s1 as slope and i as intercept (dashed gray line) for the model with an intercept. $R^2_{xy}$ is the square of the Pearson correlation coefficient. 31 samples were collected for analysis during the campaign.**



The proportionality parameter $b$ (Eq. 3) is compared with values taken from the literature in Table 3. These values depend on the location, the nature of the aerosol, and the thermal protocol used for analysis. The value of 0.44 which we determined in this study for an urban background site is slightly lower than values for other urban and urban background sites using EUSAAR

5    2 thermal protocol, and considerably lower than the values for rural sites. This shows that the determination of this parameter needs to be performed for each location and with filters sampled over the time-period of interest.

| $b$ | Thermal Protocol | Location | Reference |
|---|---|---|---|
| 0.52 | NIOSH | Fresno, CA, USA | (Chow et al., 2009) |
| 0.67 | NIOSH | Boston, MA | (Kang et al., 2010) |
| 0.30 – 0.37 | NIOSH | Rochester, Philadelphia, USA (urban) | (Jeong et al., 2004) |
| 1.27 | IMPROVE TOR | Riverside, CA, | (Babich et al., 2000) |
| 1.32 | | Chicago, IL, | |
| 1.41 | | Phoenix, AZ | |
| 1.61 | | Dallas, TX | |
| 1.59 | | Bakersfield, CA and | |
| 1.61 | | Philadelphia, PA, USA | |
| 1.64 | IMPROVE TOR | Fresno, CA, USA, winter | (Park et al., 2006) |
| 1.23 | | Fresno, CA, USA, summer | |
| 0.74 | IMPROVE TOR | Columbus, OH, USA | (Cowen et al., 2014) |
| 0.56 | IMPROVE TOT | | |
| 0.61 | Swiss_4S | Switzerland | (Zotter et al., 2017) |
| 0.54 | EUSAAR_2 | Madrid, Spain (urban) | (Becerril-Valle et al., 2017) |
| 1.23 | | Villaneuva, Spain (rural) | |
| 0.67 – 0.91 | EUSAAR_2 | Vallée de l'Arve, France (rural, woodsmoke dominated) | (Chevrier, 2016) |
| 0.96 | EUSAAR_2 | Grenoble, France (urban, woodsmoke dominated) | (Favez et al., 2010) |
| 0.88 | EUSAAR_2 | Paris, France (regional background) | (Petit et al., 2015) |
| 0.94 | EUSAAR_2 | Paris, France (regional background) | (Zhang et al., 2019) |
| 0.83 | EUSAAR_2 | Granada, Spain (urban background) | (Titos et al., 2017) |
| 0.64 | EUSAAR_2 | Vavihill, Sweden (rural background) | (Martinsson et al., 2017) |
| 0.44 | EUSAAR_2 | Ljubljana, Slovenia | *This study* |

10    **Table 3: Summary of $b$ values (Eq. 3, Eq. 9), where EC was determined by performing thermal-optical analysis (NIOSH, IMPROVE TOT, IMPROVE TOR, SWISS_4S and EUSAAR_2) on 24 h filters, while BC was measured by Aethalometer.**

Uncertainties associated with the reported Aethalometer BC mass concentrations incorporate the uncertainty in flow

15    calibration, the uncertainty in the attenuation measurement and the uncertainty in the conversion of the attenuation coefficient


to mass concentrations - constant mass attenuation cross-section approximation ( Gundel et al., 1984; Hansen, 2007, Drinovec et al., 2015, Healy et al., 2017, Zotter et al., 2017). The overall estimated uncertainty for reported BC mass concentrations is approximately 25% (World Meteorological Organization and Global Atmosphere Watch, 2016). The EC data used in the comparison depends greatly on the thermal protocol used (Karanasiou et al., 2015). In addition, the uncertainty can be

determined using the procedure described in the standard EN16909:2017. The uncertainty we use has been taken as the laboratory-to-laboratory variability of 10%.

### 3.5  Comparison of online OC measurements from TCA with offline OC filter analyses

Online OC measurements can be derived using the above EC-BC correlation plot to assign the appropriate operational value

of the parameter $b$; the online BC data; and the online TCA data. Figure 8 shows the correlation between online $OC_{TC\text{-}BC}$ and offline OC derived from the 24-hour filter samples analyzed with a thermo-optical OC-EC analyzer. These results show that when using an appropriate value of $b$, the "TC – BC Method" yields online data for the OC content of ambient aerosols that agree very well with conventional offline thermal analyses. The offset $i = 1.33 \pm 0.18$ μg/m$^3$ lies in the same range as that determined by TC correlation analysis, which confirms that organic carbon is the origin of the offset in the correlation plots in

Figs. 6 and 8. The in-depth analysis of the relative difference between OC from 24 h filters and OC determined by online measurement as TC-$b$BC shown in Fig. 9 reveals that the positive artefact can be the dominant apparent source of OC for days with very low OC concentrations (< 5 μg/m$^3$) in comparison to offline 24 h filters, for which also negative artefact (desorption of VOCs) can occur. Longer sample time base or usage of two denuder monoliths in TCA08 would decrease the offset in such comparisons. Nevertheless, as the offset lies in the same range as that determined by the inter-laboratory comparison of off-

line filter analyses (Table 2, Fig 4.), we can assume that it can be neglected and the regression without intercept can be used for intercomparison,

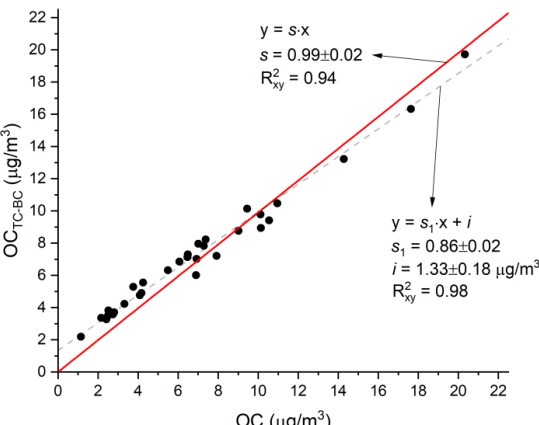

**Figure 8: Comparisons of offline measurements of OC (laboratory filter analysis) using the EUSAAR_2 thermal protocol, to the 24-hour average of online measurement of OC=TC-$b$BC data taken by the AE33 Aethalometer and TCA08 Total Carbon Analyzer. Linear orthogonal regression results (n=31) are shown with s as the slope (red line) for the model without an intercept and with s1 as slope and i as intercept (dashed gray line) for the model with an intercept. R$^2_{xy}$ is the square of the Pearson correlation coefficient.**


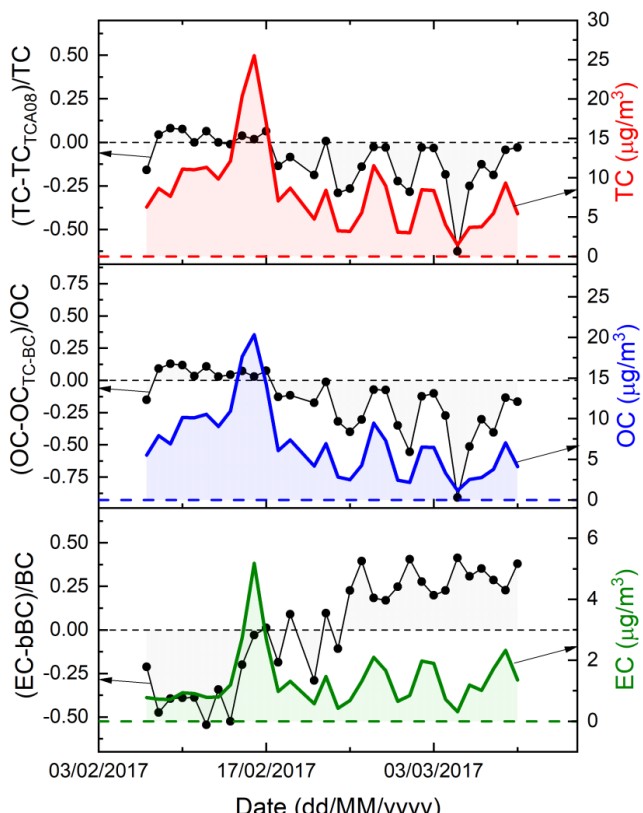

**Figure 9: Left y-axis:** Relative difference between TC, OC and EC (see Eq. 7) measured on 24 h filters by conventional OC-EC method and TC$_{TCA08}$, TC-bBC and bBC measured online by TCA08 on 1 h time resolution and AE33 on 1 min time resolution and then averaged on 24 h. **Right y-axis:** The absolute concentrations of TC, OC and EC (red, blue and green line, respectively) is shown

5    for easier comparison.

### 3.5  Comparison of OM online measurements from ACSM with offline OC from filter sampling and online OC$_{TC\text{-}BC}$

The data from an AE33 and TCA08 can be combined with an operational timebase of 1 hour, yielding OC and EC data with much greater time resolution than what can be achieved by the analysis of filter samples. In order to assess the high-time

10   resolution performance of this on-line technique, comparison of BC (from AE33) and TC (from TCA08) together with OM analyzed by ACSM is shown in Fig. 10. Due to variability in ACSM timings, the data was gathered into 3h averages. The chemical composition dependent collection efficiency of the Q-ACSM was calculated according to Middlebrook et al., 2012.





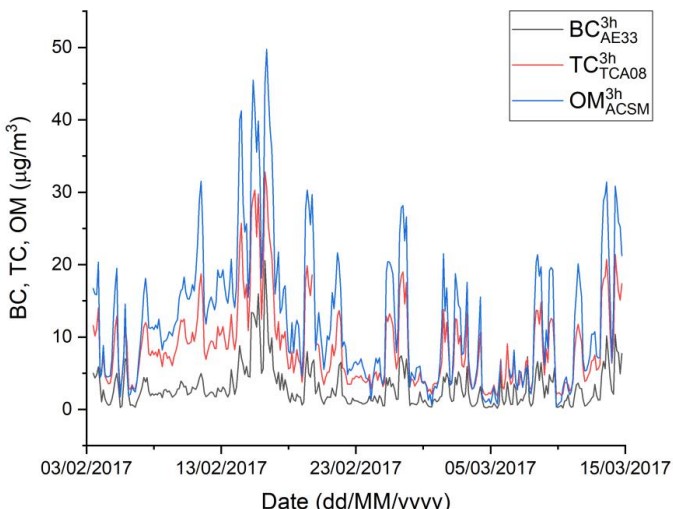

**Figure 10: Time series comparisons of high-time resolution online measurements of OM by ACSM on 29-30 min time base, BC by AE33 on 1 min time base, and TC by TCA08 on 1h time base. All data is averaged to 3h for easier comparison.**

Ambient organic-mass-to-organic-carbon ratio (OM/OC) in organic aerosol (OA) in organic aerosol (OA) is an important parameter to investigate OA chemical composition. OM/OC can vary widely depending on the sources, monitoring location, season and meteorology. The lower ambient OM/OC ratios are consistent with fresh aerosol emission from traffic, while the higher values are usually observed for aged ambient oxygenated OA (Chirico et al., 2010).

The slopes $s$ of the regressions without intercept represent average OM/OC values measured during this campaign (Fig. 11). The ratios determined from comparison of daily averages of OM measurements to OC from offline filters (Fig. 11 (a)) and to eOC from TC-$b$BC method (Fig. 11 (b)) are 1.79 and 1.82 respectively. The ratio lies on the higher end of OM to OC range determined for urban environments which is 1.4 to 1.8, while for the rural sites it varies from 1.7 to 2.3 3 (Aiken et al., 2008; Gilardoni et al., 2006; Sun et al., 2009; Turpin and Lim, 2001). This is consistent with other studies in similar urban
environments with close proximity of the sampling site to fresh vehicle emissions and additional contribution of biomass burning (Brown et al., 2013; Turpin and Lim, 2001; Xing et al., 2013). The sampling site used in this study is mainly influenced by fresh emissions from traffic with a regionally homogeneous contribution of biomass burning for household heating (Ogrin et al., 2016). The in-depth source apportionment analysis of OA and high time resolution of OM/OC ratio from this campaign will be discussed in a different study.

The negative offset in the regression model with intercept (Fig. 11 (b)) again reveals the pronounced positive sampling artefact due to adsorption of organics on quartz fiber filters for short sampling times in TCA08 method. This is not the case of the non-filter based ACSM measurement of organic aerosol mass. The influence of such sampling artefact is noticeable only during conditions with low atmospheric loading of particulate organic aerosols. Again, the installation of two denuders monoliths or
increased sample time base for TCA08 is recommended in such environment in order to minimize the influence of these sampling artefacts.





(a)

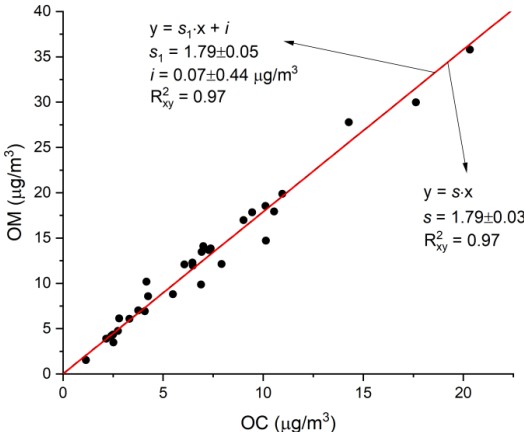

5 (b)

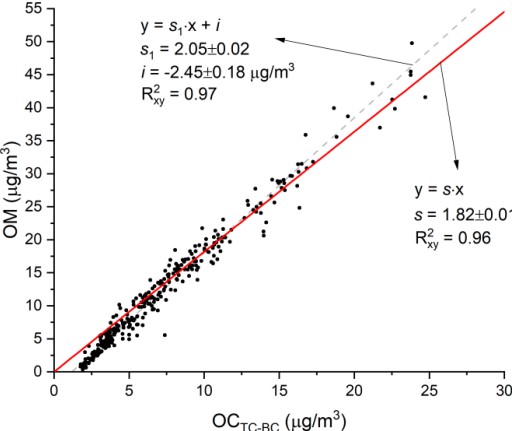

**Figure 11: (a) Comparison of offline measurements of OC (laboratory filter analysis) using the EUSAAR_2 thermal protocol, to the 24-hour average of online measurement of OM data taken by the ACSM. A total of 31 filter samples were collected for analysis**
10 **during the campaign. (b) Comparison of 3h OC data derived as OC = TC - $b$BC, to OM data measured by ACSM. Linear orthogonal regression results are shown with s as the slope (red line) for the model without an intercept and with s1 as slope and i as intercept (dashed gray line) for the model with a intercept. $R^2_{xy}$ is the square of the Pearson correlation coefficient. 300 data points are used in the regression analysis.**



### 3.6 Diurnal profiles of high-time resolution measurements of eOC, eEC, and eEC/TC ratio

The coupling of TCA08 and Aethalometer instruments offers new opportunities to investigate the short-term variability of carbonaceous aerosols, and the factors that control their atmospheric concentrations such as source variability and/or atmospheric (dynamic/photochemical) processes. For this purpose, diurnal profiles of organic carbon and elemental carbon concentrations were calculated for each hour of the day (Fig. 12 (a)), separately grouped for working days (Monday to Friday) and for weekends (Saturday and Sunday). The diurnal variation of OC and EC for this urban background environment is strongly influenced by the temporal patterns of emissions from traffic and biomass burning (domestic heating) during wintertime. Two traffic peaks can be observed for working days in OC and EC concentrations; the first one observed during morning rush hours (between 6:00 and 10:00 LT) and the second in the afternoon, after 16:00 LT. Between the two peaks, (e.g. between 10:00 and 16:00), OC and EC concentrations decrease due to atmospheric dilution in the increasing mixing height of the planetary boundary layer (Ogrin et al., 2016). During the weekend the morning traffic peak disappears, while the evening one remains present. Peaks in average eEC to average TC ratio are concomitant with the eEC peaks which is aligned with the EC-rich pattern of traffic emissions (Fig 12 (b)). Average eOC and eEC values during the measurement camping were $7.3 \pm 4.9 \ \mu g/m^3$ and $1.3 \pm 1.3 \ \mu g/m^3$, respectively, which is consistent with 24h filter measurements of OC and EC at the other urban background location in Ljubljana (Biotehniška fakulteta), where averaged values for OC and EC of 8.4 and $1.0 \ \mu g/m^3$ were measured for the period between October 2016 and March 2017 (Gjerek et al., 2018).

.

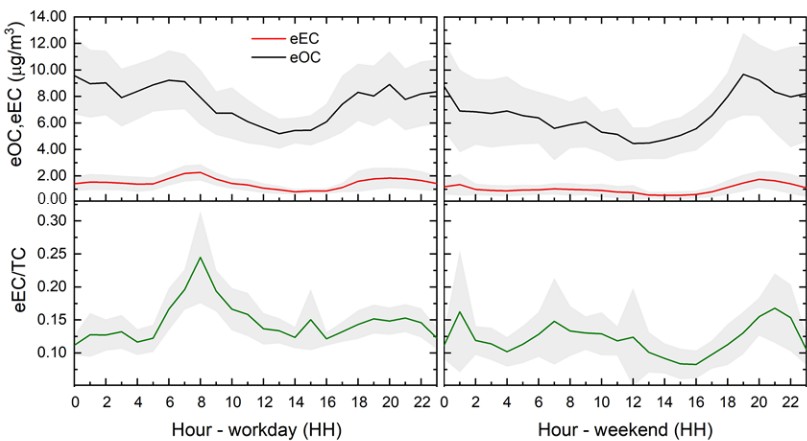

**Figure 12: Hourly diurnal profiles for workday (left) and weekend (right) for eOC (black line) and eEC (red line) and average eEC to average TC ratio (green line). The gray shaded area represents 95% confidence interval around mean value.**

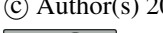



**4. Summary**

We present the newly developed Total Carbon Analyzer model TCA08, which offers measurement of the concentrations of
total aerosol carbon continuously with high time resolution as rapid as 20 min.  Two parallel flow channels provide continuous
operation: while  one channel analyzes, the other collects the next sample. Thermal analysis by flash-heating of the sample
collected on a quartz fiber filter efficiently converts all the particulate carbon to $CO_2$. The increase in $CO_2$ concentration above
baseline in a flow of analytic air is measured by an integrated NDIR detector.  When the TCA08 is combined with an
AE33Aethalometer, the TC-BC method yields OC-EC data with much greater time resolution than that offered by the analysis
of filter-based samples. In this study, we show results from these instruments combined on an operational timebase of 1 hour
and compare them to conventional 24h filter measurements of EC and OC, and high-time-resolution measurements of organic
aerosols with an ACSM. The correlation analysis showed very high agreement between for eOC = TC-$b$BC and eEC= $b$BC
derived by the TC-BC method, to OC-EC analysis using EUSAAR2 thermal protocol on 24h filters  and OM from ACSM.
The value of the calibration parameter $b$ can be derived for the desired OC-EC thermal protocol to obtain high time resolution
eOC and eEC data.

These two instruments are automatic, rugged, and designed for unattended operation in field monitoring situations.
Measurements can be done in different PM size fractions (PM$_1$, PM$_{2.5}$, PM$_{10}$)  The combined data may be analyzed to examine
repetitive diurnal patterns, reflecting both anthropogenic inputs of carbonaceous aerosols to the atmosphere; production of
secondary aerosols; as well as atmospheric processing and dispersion into mixing layers of varying depth.  Additional analyses
can compare these results between workdays and weekends, seeking patterns of human activity that may reflect changes in
traffic or industrial emissions.  Studies such as this, requiring large numbers of closely-spaced data points, are greatly facilitated
by online instruments.

**Acknowledgments**
This work was partly funded by the EUROSTARS grant E!8296 TC-BC.

**Code/Data availability**
The data used in this publication is available upon request to the corresponding author (martin.rigler@aerosol.eu).

**Author contribution**
MR, LD, and GM designed the study, MR, GL, LD, GM, and IS  performed and analysed TC, BC and OM online
measurements. MR, GL, LD, GM,  AV, AP, and AH were involved in the new instrument development. JJ, JS, JB, IK, and JT
preformed OC/EC measurements on offline filters. All authors contributed to the scientific discussion.

**Conflicts of interest**
At the time of the research, M. Rigler, L. Drinovec, G. Lavrič, and G.Močnik were also employed by the manufacturer of the
Aethalometer and Total Carbon instruments. Other authors declare no conflict of interest. The funding sponsors had no role in
the design of the study; in the collection, analyses, or interpretation of data; in the writing of the manuscript, and in the decision
to publish the results.



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
