# Peer review of "The new instrument using TC-BC method for the online measurement of carbonaceous aerosols"

_Atmospheric Measurement Techniques, 2019_

## Referee Comment (RC1) · Anonymous Referee #1 · 30 Dec 2019

The paper describes a new type of analyzer that can be coupled with the AE33 Aethalometer to provide Total Carbon measurements and estimate equivalent Organic and Elemental carbon concentrations in particulate matter deposited on a filter. While a field analyzer offering OC/EC/TC measurements has already been available in the market, this new instrument incorporates several novelties, mostly in design, like the use of a dual sampling channel, application of zero air as carrier gas and durability of materials. The instrument and method can potentially benefit AE33 users and help them obtain additional TC, eOC and eEC results. It can further compliment the offline OC/EC reference method with high-time resolution measurements. As with all new analyzers, it should go through type testing and validation, which is the main weakness of the current paper since this is partially dealt with.

Comments:

As mentioned on the Title, page 1 line 16, page 4 line 1 and elsewhere the authors state that this is a new or newly developed method. Nevertheless, a method bearing great similarities has been already described in the past (Bauer et al, 2009 and references therein). The paper describes an analyser of a different manufacturer that "also determines optical OC and optical EC by monitoring the laser transmission data through the quartz filter", "Total carbon (TC) is determined using the thermal-optical method, and then optical OC is deduced by subtracting optical EC from TC (optical OC = TC - optical EC)". Where optical OC and EC would be simply a different terminology given for eOC and eEC used in the current paper. How would the authors comment on the method similarities of the two studies and the suitability of the description "new" for the method?

While the terms of OC, EC and eOC, eEC are clearly defined, their use in the text overlaps and is occasionally confused. Proper terminology should be consistently used in order to avoid any misinterpretations by readers. For example, the abstract mentions in lines 22-23 that this new application can result in high time resolution determination of organic and elemental carbon while in reality it provides an estimation of eOC and eEC values. Another example would be in section 3.6: eOC and eEC should be used instead of OC and EC. Also applicable in all graphs.

NDIR detectors, similarly to the one in the current application, may deteriorate in performance in long term and show a drift in their baseline. Since there is no application of an internal standard calibration or a span check, have the authors evaluated how often would an external standard calibration be required? Would there be any NDIR detector related maintenance needs, e.g. source replacement, and in what frequency would that be required?

The last paragraph of section 2.3 describes tests performed on the denuder efficiency but it seems that results are not included in the paper. Page 7, line 21 also refers to TC data "(see below", which are not presented later on in the text. Related to the denuder efficiency, there is clear evidence of a positive artefact for eOC concentrations below 8μg/m3, visible in Figure 8 (OC vs eOC) as well as in Figure 11 (b) (OM vs eOC). Further there are signs of a negative artefact for higher concentrations, based on the same graphs, suggesting reduced combustion efficiency. The later would be more profound if there would be an addition of second denuder monolith as suggested in page 12 lines 25-27 or if a correction for the positive artefact would be applied. Should the user then consider the use of 2 correction factors (b)

related to the concentration levels measured? Or would there be any other suggestion to overcome these issues?

Section 3 refers to EN 16450:2017 regarding the orthogonal regression analysis on the 31 daily measurements between the new instrument (candidate method) and 2 independent laboratories (reference method). A proper application of EN16450:2017 would require a minimum of 40 valid data pairs with the further requirement of 2 candidate applications for each type testing application. EN16450:2017 further describes requirements related to the number of locations and the concentration range of data points. The use of just one candidate method limits the conclusions on performance consistency between identical instruments and restricts the candidate method uncertainty calculations. It seems that quite a significant part of section 3 discusses the comparability between the two reference method applications, which is a topic thoroughly documented elsewhere in the literature (Inter-comparison exercises publications are included in the reference list of the current paper). A more relevant approach would have included and compared two new instruments in parallel measurements. How would the authors comment on the approach applied and the data suitability?

The uncertainty limit value of 2.00 µg/m$^3$ mentioned in page 10 line 13 originates from calculations of PM reference methods where limit values of 30µg/m$^2$ or more apply. Would that be directly applicable for TC method and concentrations? Further section 3.3 describes method uncertainty calculations based on the NDIR detector response. Have the authors considered additional sources of uncertainty to be included in the uncertainty budget, e.g. use of denuder, zero air carrier gas, ambient temperature and pressure variations?

High volume samples were collected for analysis by the two laboratories. Is there information available on the type of the filters used? Was the homogeneity of the samples estimated for this study? Did the laboratories analyze the samples in triplicate or duplicate and were there standard external solutions analyzed like in most of the comparison exercises referenced in the text? Standard solutions may provide an insight if the difference between the TC results of the 2 laboratories was a result of calibration deviations.

A common practice in comparison exercises is following identical procedures on filter handling, transport and storage for all participants. In the current case filters were first analyzed by the ARSO laboratory and then shipped to IGE for further analysis. Even though the authors mention that "sampling, transport and storage of the filters was done according to the EN 16909:2017" IGE received the filters for analysis at a later period, after additional transport, handling and storage. Could that have contributed to the small uncertainty observed between the reference methods? Wouldn't an approach of dividing the samples and shipping in both labs in parallel have resulted in improved comparability of the two laboratories?

Table 3 provides with a wide range of b factors and the recommendation right above is: "the determination needs to be performed for each location and with filters sampled over the time-period of interest". Considering that the b factor is location and season specific, could the authors suggest a typical coverage period range with OC/EC offline analysis in parallel? How often would the b factor have to be re-evaluated per location?

The calculated b factor for this study (0.44) is the lowest among EUSAAR_2 users of the literature listed in Table 3. Further the slope from the OM – eOC comparison of (Figure 11,(b)) is 1.82, and would have

been even higher when considering the high negative intercept. Following the ranges provided by the literature in page 17, lines 14-15, these slopes fit better a rural site rather than an urban environment. This comes in contradiction with the characteristics of the selected site which is influenced from traffic emissions. If the estimated b factor was within the literature range that would have further resulted in a lower OM - eOC slope and would fit the literature range better. How would the authors comment on these observed differences compared to the literature?

Following Graph 7, it seems that in the first half of the campaign BC is overestimated while on the second half underestimated, compared to EC. Would there be any interpretations for this observation? Further, around the 5th of March eOC and TC from the TCA08 configuration are overestimated and bBC is underestimated, significantly more from the rest of the data points. Would there be any justification by the authors?

Page 15, lines 19-21, suggest that the intercept due to the eOC positive artefact can be neglected based on the comparability with the OC intercept of the two independent laboratory measurements. Nevertheless, as discussed earlier in the text, the eOC intercept is systematic and attributed to the denuder performance. It should also be noted that the eOC concentrations are compared to the average values between the two laboratories. It would be more appropriate if the artefacts observed for a new instrument would not be overlooked but rather investigated further and be dealt with. The use of two candidate method analyzers and an extended data set would be required for in-depth analysis. The above comes also in contradiction with the conclusion of page 20, line 12: "the correlation analysis showed very high agreement between eOC and eEC to the EC and OC". It seems that there is room for improvement in agreement once the artefact issues are resolved.

Technical corrections:

Page 2, line 26: tTe.

Page 15, line 4: It is not clear on which comparison the authors refer.

Page 17, line 21: Fig 10(b) to Fig 11(b).

Page 18: Figure 11 (a) misses the dashed trendline.

References
Jace J. Bauer, Xiao-Ying Yu, Robert Cary, Nels Laulainen & Carl Berkowitz (2009) Characterization of the Sunset Semi-Continuous Carbon Aerosol Analyzer, Journal of the Air & Waste Management Association, 59:7, 826-833, DOI: 10.3155/1047-3289.59.7.826

---

## Referee Comment (RC2) · Anonymous Referee #2 · 31 Dec 2019

**Review Comments on AMT-2019-376**

*General Comments*

The manuscript presented a new system for continuous monitoring of carbon fractions in ambient particle samples and conducted comparisons between the online method with the offline filter based thermo-optical OC-EC method. The advantages of the new system, i.e. easy-to-deploy and low maintenance, are convincing while a few details regarding data treatment and analytical specifications need to be elaborated before the manuscript can be considered for publication.

*Specific Comments*

Introduction:

The intro part mainly discussed the definitions of different carbon fractions and various protocols for measuring the carbon fractions. However, some of the content is repetitive (e.g. it was discussed in Page 2 Line 26 to Page 3 Line 2 that a few factors will influence the OC-EC split while similar points were mentioned again in Page 3 Lines 9–27). The authors also listed out three protocols that have been widely used in different regions of the world (Page 2 Lines 16–24). Since in the following sections the online data were compared with the offline data obtained from the EUSAAR2 protocol, readers might expect to see more discussions on the specific differences among the protocols (EUSAAR2 vs. IMPROVE_A, EUSAAR2 vs. NIOSH).

Section 2.2:

How is the performance of external calibration (using sucrose or KHP or other chemicals) by TCA08? What is the maximum carbon concentration tested?

Section 3.3:

What is the "carbon calibration factor"? How is it derived?

What's the loading effect compensation algorithm used for treating AE33 data in this study? Will different algorithms introduce uncertainties?

Section 3.4:

It can be seen from this work and from previous literature data (Table 3) that the relationship between BC and EC is location dependent. If the aerosol composition at a certain location has a very clear temporal variation pattern, the parameter $b$ could be sensitive to the sampling time period as well. Since the PM monitoring networks usually adopt the filter-based sampling approach followed by offline laboratory analysis and the historical dataset was very likely obtained from offline measurement, do the authors suggest that every time the online TC-BC system is deployed to one sampling location, the online-offline comparison needs to be conducted to derive the $b$ value so that the measured data can be compared to other dataset?

---

## Author Comment (AC1) · 27 Feb 2020

**Point-to-point response to Referee #1 (RC1)**

We are grateful for yours comments on how to strengthen our manuscript. Please find attached revised manuscript and see our point-to-point response to your comments below:

1. **As mentioned on the Title, page 1 line 16, page 4 line 1 and elsewhere the authors state that this is a new or newly developed method. Nevertheless, a method bearing great similarities has been already described in the past (Bauer et al, 2009 and references therein). The paper describes an analyser of a different manufacturer that "also determines optical OC and optical EC by monitoring the laser transmission data through the quartz filter", "Total carbon (TC) is determined using the thermal-optical method, and then optical OC is deduced by subtracting optical EC from TC (optical OC =TC -optical EC)".Where optical OC and EC would be simply a different terminology given for eOC and eEC used in the current paper. How would the authors comment on the method similarities of the two studies and the suitability of the description "new" for the method?**

There are conceptual similarities between method mentioned above (Bauer et al., 2009) and TC- BC method developed in this study: the new method takes the advantage of decoupling thermal and optical method into two separate instruments, both dedicated for different measurements. With this, the new method has higher time resolution, no dead time, online loading compensation for eBC measurements and is more convenient for field measurements as the thermal measurement is done without fragile quartz cross oven, high purity gases and catalyst. The main difference are listed and described in details below:

- The optical EC in semi continuous Sunset instrument is a measurement of transmittance through the filter at a wavelength of 660 nm prior to the thermal analysis, while the equivalent eEC is an equivalent BC measurement with AE33 at 880 nm, then multiplied by a proportionality factor $b$ (eEC = $b$BC). Contribution of light absorbing organics is higher at 660 nm.
- Light source used in AE33 is non-coherent set of light emitting diodes (LEDs) with diffuser-like optics (BC6 is measured at 880 nm), while the semi continuous Sunset instrument uses diode laser at 660 nm. Distorted wave fronts of LEDs produce homogeneous signal on the exposed filter area and the transmittance signal has no speckle/interference noise, making the LED light source far more convenient for filter attenuation (ATN) measurements.
- The presence of the loading effect in filter-based absorption photometers causes an ATN-dependent change in the instrumental sensitivity. The Aethalometer model AE33 performs real time compensation for this nonlinearity using patented dual spot algorithm.
- ATN in AE33 is determined more precisely as detector intensity signal for the measurement spot $I$ and detector signal for the reference spot $I_o$ are measured concurrently. Any drift in LED light intensity due to temperature and other changes are compensated in real time. While the stabilization of laser source is possible but there are no publication describing this process for Sunset semi-continuous instrument.
- When the attenuation reaches a certain threshold, a tape advance is induced so that measurements starts on a clean spot. Significant buildup of refractory substances on

the filter in the semi continuous Sunset instrument will reduce initial intensity and introduce higher signal to noise ratio at each subsequent optical analysis. Two dedicated instruments are better for this purpose.

A short paragraph was added to the manuscript (Chapter: 2.1 TC-BC method for online high time resolved OC-EC measurements, p. 4, lines 17-21:

- Although one can find conceptual similarities between method presented in Bauer et al., 2009 (and references therein) and TC- BC method presented in this study, the new method takes the advantage of decoupling thermal and optical method into two separate instruments, both dedicated for different measurements. With this, the new method has higher time resolution, no sampling dead time, online loading nonlinearity compensation for eBC measurements (Drinovec et al., 2017) and is more convenient for field measurements as the thermal measurement is done without fragile quartz cross oven, high purity gases and catalyst.

Two references were added to the manuscript:

- Bauer, J. J., Yu, X.-Y., Cary, R., Laulainen, N., and Berkowitz, C.: Characterization of the Sunset Semi-Continuous Carbon Aerosol Analyzer, J. Air Waste Manag. Assoc., 59(7), 826–833, doi:10.3155/1047-3289.59.7.826, 2009.

- Drinovec, L., Gregorič, A., Zotter, P., Wolf, R., Bruns, E. A., Prévôt, A. S. H., Petit, J.-E., Favez, O., Sciare, J., Arnold, I. J., Chakrabarty, R. K., Moosmüller, H., Filep, A., and Močnik, G.: The filter-loading effect by ambient aerosols in filter absorption photometers depends on the coating of the sampled particles, Atm. Meas. Tech., 10(3), 1043–1059, doi:10.5194/amt-10-1043-2017, 2017.

2. **While the terms of OC, EC and eOC, eEC are clearly defined, their use in the text overlaps and is occasionally confused. Proper terminology should be consistently used in order to avoid any misinterpretations by readers. For example, the abstract mentions in lines 22-23 that this new application can result in high time resolution determination of organic and elemental carbon while in reality it provides an estimation of eOC and eEC values. Another example would be in section 3.6: eOC and eEC should be used instead of OC and EC. Also applicable in all graphs.**

Terms OC, EC and eOC and eEC where changed throughout the manuscript using proper terminology as follows:
- Page 1, line 20: The concentration of particulate equivalent organic carbon (eOC) is determined by subtracting black carbon concentration, concurrently measured optically by an Aethalometer®, from the total carbon concentration measured by the TCA08.
- Page 1, line 23: The combination of TCA08 and Aethalometer (AE33) is an easy-to-deploy and low maintenance continuous measurement technique for the high time resolution determination of equivalent organic and elemental carbon (eEC) in different particulate matter size fractions, which avoids pyrolytic correction and need for high purity compressed gases.
- Page 9, Table 2:

| OC (see Eq.7) | eOC | 31 | 0.94 | 0.99 ± 0.02 | 0.98 | 0.86 ± 0.02 | 1.33 ± 0.18 |
| eOC | OM$_{ACSM}$ | 300 | 0.96 | 1.82 ± 0.01 | 0.97 | 2.05 ± 0.02 | -2.45 ± 0.20 |

Table 2: Summarized comparison results between off-line filter measurements and 24 h average values of high-time resolution measurements of TC, BC, eOC and OM; and between high time resolution measurements (3h) of eOC and OMACSM measurements.

- Page 16, line 15: 3.5Comparison of online eOC measurements from TCA with offline OC filter analyses.
- Page 16, line 16: Online eOC measurements can be derived using the above EC-BC correlation plot to assign the appropriate operational value of the parameter *b*; the online BC data; and the online TCA data.
- Page 16, line 19: These results show that when using an appropriate value of *b*, the "TC – BC Method" yields online data for the eOC content of ambient aerosols that agree very well with conventional offline thermal analyses.
- Page 16, line 23: The in-depth analysis of the relative difference between OC from 24 h filters and eOC determined by online measurement as TC-*b*BC shown in Fig. 9 reveals that the positive artefact can be the dominant apparent source of OC for days with very low OC concentrations ($< 5\ \mu g/m^3$) in comparison to offline 24 h filters, for which also negative artefact (desorption of VOCs) can occur.
- Page 17, Figure 8. eOC on y-axis
- Page 18, Figure 9. eOC and eEC on y-axis.
- Page 18, line 8: 3.6 Comparison of OM online measurements from ACSM with offline OC from filter sampling and online eOC
- Page 18, line 9: The data from an AE33 and TCA08 can be combined with an operational timebase of 1 hour, yielding eOC and eEC data with much greater time resolution than what can be achieved by the analysis of filter samples.
- Page 20, Figure 11: eOC on x-axis.
- Page 20, Line 10: (b) Comparison of 3h eOC data derived as eOC = TC - bBC, to OM data measured by ACSM.
- Page 21, line 6: The diurnal variation of eOC and eEC for this urban background environment is strongly influenced by the temporal patterns of emissions from traffic and biomass burning (domestic heating) during wintertime.
- Page 21, line 9: When the TCA08 is combined with an AE33Aethalometer, the TC-BC method yields eOC-eEC data with much greater time resolution than that offered by the analysis of filter-based samples.

3. **NDIR detectors, similarly to the one in the current application, may deteriorate in performance in long term and show a drift in their baseline. Since there is no application of an internal standard calibration or a span check, have the authors evaluated how often would an external standard calibration be required? Would there be any NDIR detector related maintenance needs, e.g. source replacement, and in what frequency would that be required?**

Light source life in Licor 840A NDIR detector is estimated to be 18000 hours. When light source fails the TCA instrument detects it, stops the measurements and displays Licor CO2 Error status.

Total Carbon content of the sample measured by TCA08 is a function of a CO2 difference between signal and background values and thus not directly connected to absolute value of CO2 (Eq. 4). That is why the TC result is less dependent on the light source drift in the NDIR

detector than if the absolute value is used in the calculations. Additionally, the drift of the dual wavelength light source in the NDIR detector used in TCA08 is compensated online. Concentration measurements of $CO_2$ are based on the difference ratio in IR absorption between sample and reference signal. The $CO_2$ sample uses an optical filter centred at wavelength of 4.26 μm corresponding to an absorption band for CO2. The reference channel for CO2 has an optical filter centered at 3.95 μm, which has no absorption due to CO2.

During NDIR detector lifetime there is no need to preform internal standard calibration and span check for TC measurements, as the whole system (NDIR detector + TCA analytic chamber) can be calibrated or validated with Carbon Calibration and Carbon Validation procedure for TCA08. This the great benefit of this instrument. Both procedures are described in TCA08 User Manual (TCA08, 2019). Carbon calibration of TCA08 should be done once per year or after any major maintenance or modification of the system.

A short paragraph was added to the manuscript (p. 6, lines: 11-20):

- Light source life in LI-840A CO2/H2O Analyser is estimated to be 18000 hours. When light source fails the TCA instrument detects it, stops the measurements and displays Licor $CO_2$ error status. Total Carbon content of the sample measured by TCA08 is a function of a $CO_2$ difference between signal and background values and thus not directly connected to absolute value of $CO_2$ (Eq. 4). This is why the TC result is less dependent on the light source drift in the NDIR detector than if the absolute value is used in the calculations. During light source lifetime there is no need to preform internal standard calibration and span check for NDIR detector, as the whole system (NDIR detector + TCA08 analytic chamber) can be calibrated or validated with Carbon Calibration and Carbon Validation procedure for TCA08, which is the great benefit of this instrument. Both procedures are described in TCA08 User Manual (TCA08, 2019). Carbon calibration of TCA08 should be done once per year or after any major maintenance or modification of the system.

4. **The last paragraph of section 2.3 describes tests performed on the denuder efficiency but it seems that results are not included in the paper. Page 7, line 21 also refers to TC data "(see below", which are not presented later on in the text. Related to the denuder efficiency, there is clear evidence of a positive artefact for eOC concentrations below 8μg/m3, visible in Figure 8 (OC vs eOC) as well as in Figure 11 (b) (OM vs eOC). Further there are signs of a negative artefact for higher concentrations, based on the same graphs, suggesting reduced combustion efficiency. The later would be more profound if there would be an addition of second denuder monolith as suggested in page 12 lines 25-27 or if a correction for the positive artefact would be applied. Should the user then consider the use of 2 correction factors (b) related to the concentration levels measured? Or would there be any other suggestion to overcome these issues?**

As we noted in the first paragraph of chapter 2.3 the measurement of carbonaceous aerosols using quartz-fiber filters is always very challenging because of the possibility of positive and negative sampling artifacts which are hard to quantify. The TCA08 instrument was developed in a way, that sampling face velocity was similar to the one of high volume samplers for offline analysis for easier comparison. With shorter sample time basis the positive artefact is much more pronounced and negative artefact can be neglected.

We believe that TCA08 with its denuder efficiency procedure is a great tool to investigate (1) denuder efficiency and (2) temporally variations of positive/negative artefact. In our current study (Gregorič et al., 2019, 2020), positive and negative artefact on quartz filters depending on sample time base, face velocity, number of denuders, chemical deposition of aerosol is investigated in details. When investigating dependence on sample time base, we found out that adsorption of organic vapors in TCA08 can be described with a sum of two exponential functions with $a1*(1 - Exp[-x/τ1]) + a2*(1 - Exp[-x/τ2]$ with a fast and slow time constants $τ1$ and $τ2$. Slow time constants are around 15-60 minutes, indicating that positive artefact prevails for sampling times up to 2 hours.

Nevertheless, high VOC concentrations are usually connected to high OC concentrations as well. We believe that negative artefact due to combustion efficiency mentioned by reviewer is not the reason for lower slope. External calibration of TCA shows that combustion efficiency does not reduce with higher TC, OC or EC concentrations (see Figure 1), therefore, two correction factors are not needed. Users should use appropriate number of denuders and choose appropriate sampling timebase according to denuder efficiency test and ambient TC and OC concentrations. In our next publication guidelines and recommendations on how to asses positive/negative artefact with TCA08 and how to use this knowledge for more quality measurement data will be described.

[Figure]

*Figure 1 External calibration of TCA08 - TC mass calibration range*

The last paragraph in section 2.3 was changed accordingly:
- We developed these routines during the instrument design and performed the measurements as part of the field campaign . After five weeks of continuous operation with consistent TC data , the measured denuder efficiency was 74%. We recommend that the denuder should be replaced or regenerated when its efficiency drops below 70% (Ania et al., 2005; Bhagawan et al., 2015; Gao et al., 2014). The Standard Operating Procedure for routine use of the TCA08 instrument recommends replacement or regeneration of the denuder honeycomb element once per month. Further, in environments with high VOC concentrations, two denuder honeycombs in series are recommended (Gregorič et al., 2020).

5. **Section 3 refers to EN 16450:2017 regarding the orthogonal regression analysis on the 31 daily measurements between the new instrument (candidate method) and 2 independent laboratories (reference method). A proper application of EN16450:2017 would require a minimum of 40 valid data pairs with the further requirement of 2 candidate applications for each type testing application. EN16450:2017 further describes requirements related to the number of locations and the concentration range of data points. The use of just one candidate method limits the conclusions on performance consistency between identical instruments and restricts the candidate method uncertainty calculations. It seems that quite a significant part of section 3 discusses the comparability between the two reference method applications, which is a topic thoroughly documented elsewhere in the literature (Intercomparison exercises publications are included in the reference list of the current paper). A more relevant approach would have included and compared two new instruments in parallel measurements. How would the authors comment on the approach applied and the data suitability?**

We agree with reviewer that equivalence comparison was not done exactly according to standard EN 16450:2017, but it is our best attempt to do it so. As there is no standard for reference method for online measurement of OC and EC concentrations available at the time of the writing of this manuscript, we followed EN16450:2017 and choose EN 16909:2017, 2017 as the reference method. The comparison was done on the available data (31 daily filters - limitations of DIGITEL high volume sampler availability). Furthermore, we used only one set of instrument for the candidate method comparison, as they are both compared to the reference set of instruments after their assembly (in-house defined requirements for successful intercomparing test are: 1. TCA08: TC concentration range up to 75.000 ng/m$^3$, slope between 0.97-1.03, $R^2$ above 0.98 ; 2. AE33: eBC concentrations up to 30.000 ng/m$^3$; slope between 0.97-1.03, $R^2$ above 0.98). Table below (Table 1) shows results of such comparison for TCA08 instrument.

|  | Instruments serial number | Slope | $R^2$ | N | TC value max (ng) | TC value min (ng) |
|---|---|---|---|---|---|---|
| 1 | TCA08-S00-0131 | 1.0216 | 0.9975 | 65 | 45856 | 2732 |
| 2 | TCA08-S00-0132 | 1.0098 | 0.9992 | 65 | 45803 | 2732 |
| 3 | TCA08-S00-0133 | 1.0041 | 0.9993 | 19 | 67274 | 6286 |
| 4 | TCA08-S00-0134 | 0.9987 | 0.9989 | 19 | 67274 | 6286 |
| 5 | TCA08-S00-0135 | 0.9967 | 0.9994 | 19 | 67274 | 6286 |
| 6 | TCA08-S00-0136 | 0.9874 | 0.9992 | 139 | 76684 | 2498 |
| 7 | TCA08-S00-0137 | 0.9964 | 0.9987 | 65 | 51119 | 2592 |
| 8 | TCA08-S00-0138 | 0.9867 | 0.9993 | 65 | 51119 | 2592 |
| 9 | TCA08-S00-0139 | 0.9605 | 0.9988 | 65 | 51119 | 2592 |
| 10 | TCA08-S00-0140 | 1.0168 | 0.999 | 62 | 73190 | 2644 |
| 11 | TCA08-S00-0141 | 0.9974 | 0.9998 | 22 | 64579 | 3336 |
| 12 | TCA08-S00-0142 | 1.0057 | 0.998 | 136 | 76466 | 2823 |
| 13 | TCA08-S00-0143 | 0.9967 | 0.9989 | 22 | 64579 | 3336 |
| 14 | TCA08-S00-0144 | 0.9972 | 0.9993 | 22 | 64609 | 3336 |
| 15 | TCA08-S00-0145 | 0.9699 | 0.9978 | 137 | 72992 | 2823 |

*Table 1 Slope mean value for the intercomparison of 15 TCA08 instruments (SN from S00-0131 to S00-0145) to the reference instrument in-house (SN S00-0103) is 0.996 ± 0.015 with min R2 of 0.9975. The intercomparison is done on 20 min sample time base with TC concentrations up to 75000 ng/m3.*

Following paragraph was added to the manuscript (page 9, lines: 12-21):

- As there is no standard for reference method for online measurement of OC and EC concentrations available at the time of the writing of this manuscript, we followed EN16450:2017 and choose EN 16909:2017 as the reference method. Nevertheless, a proper application of EN16450:2017 would require a minimum of 40 valid data pairs with the further requirement of two candidate applications for each type testing application. Additionally, the same standard further describes requirements related to the number of locations and the concentration range of data points. The results and discussion in this chapter is our best attempt of equivalence comparison on the available data (31 daily filters due to the limited access to the DIGITEL high volume sampler). Furthermore, we used only one set of instrument for the candidate method comparison, as they are both compared to the reference set of instruments after their assembly as one of the tests during final inspection procedure (in-house defined requirements for successful intercomparison test are: 1. TCA08: TC concentration range up to 75.000 ng/m3, slope between 0.95-1.05, R2 above 0.98 ; 2. AE33: eBC concentrations up to 20.000 ng/m3; slope between 0.95-1.05, R2 above 0.98).

6. **The uncertainty limit value of 2.00 µg/m$_3$ mentioned in page 10 line 13 originates from calculations of PM reference methods where limit values of 30µg/m$_2$ or more apply. Would that be directly applicable for TC method and concentrations? Further section 3.3 describes method uncertainty calculations based on the NDIR detector response. Have the authors considered additional sources of uncertainty to be included in the uncertainty budget, e.g. use of denuder, zero air carrier gas, ambient temperature and pressure variations?**

Carbonaceous aerosols frequently account for a large and often dominant fraction of fine particulate matter (PM2.5) mass in polluted atmospheres. Total carbon mass concentrations can contribute up to 50% of PM2.5 mass in special pollution events, which justify the use of uncertainty limit value of 2.00 µg/m$^3$ for TC concentrations.

Temperature and pressure variations during sampling are measured with meteorological sensor and are included into calculations as the volumetric sample flow is used for concentration calculations. During analysis temperature and pressure of the analytic stream are measured within NDIR sensor and are included in $CO_2$ concentration determination.

Uncertainty due to variations in zero carrier (analytic) gas during thermal analysis was tested by measuring replicates of blank samples. This approach is also used for Limit of Detection determination, which is 0.3 µg of TC for TCA08. Again, before entering the chamber, the analytic air passes through a 10-liter buffer volume for ambient $CO_2$ fluctuation averaging and a capsule filter filled with activated carbon and pleated glass fiber filter, which removes organic gases and particles from the stream.

An example of such LoD determination for TCA08-S00-00103 is shown in table below (Table 2)

| Blank measurement | TC counts (ppm) | TC (ng) |
|---|---|---|
| 1 | 11.8427 | 186.4 |
| 2 | 1.3997 | 23.22 |
| 3 | 0.5914 | 9.31 |
| 4 | 2.4629 | 40.86 |
| 5 | 7.7607 | 122.15 |
| 6 | 9.803 | 162.63 |
| 7 | 0.0469 | 0.74 |
| 8 | 3.2019 | 53.12 |
| 9 | -0.7977 | -12.56 |
| 10 | 8.2248 | 136.45 |
| 11 | -5.9253 | -93.26 |
| 12 | 12.3681 | 205.19 |
| 13 | -9.4706 | -149.07 |
| 14 | 17.5429 | 291.04 |
| 15 | 9.1144 | 143.46 |
| 16 | 4.7438 | 78.7 |
| 17 | 1.0516 | 16.55 |
| 18 | -2.4038 | -37.84 |
| 19 | 6.3558 | 105.44 |
| 20 | 12.1225 | 201.11 |
| 21 | 10.3992 | 163.68 |
| 22 | 5.6442 | 88.84 |
| 23 | -3.9845 | -62.72 |

*Table 2 Measurements of replicates of blank samples in order to determine Limit od detection (LoD). LoD = mean + 2 SD = 300 ng of TC*

We did not include positive artefact and denuder efficiency in the uncertainty budget, as they are also not considered in the uncertainty budget in standards EN 12341:2014 and EN 16909:2017:

- **EN 12341:2014, Chapter 9.3.2.11:** Artefacts due to interactions between the filter material and gasses: In addition to water, filter materials my adsorb volatile compounds presented in the sampled air. Examples hereof are ammonia, nitrogen dioxide and organic gases. Contributions to the filter mass will vary with concentration of the gases and the chemical nature of the filter material. Adsorption may even lead to a reduction of losses of semi-volatile constituents of PM (9.3.2.4). Consequently, the magnitude of the effects of adsorption of gases cannot be quantified. For the purpose of application of this European Standard the phenomenon is recognized but not considered in the uncertainty budget.
- **EN 16909:2017, Chapter 9.2**: All sampling artefacts are inherent by convention and part of the EC and OC values according to this standard. Sampling artefacts are mainly to be expected for OC and they can be significant (Chow et al. [12]).

A following sentences were added to the manuscript:

- (page 11, lines 12-13):
  The uncertainty $u_{RM}$ between the reference methods for TC

$$u_{RM}^2 = \frac{1}{2n}\sum_{i=1}^{n}\left(TC_{i,ARSO} - TC_{i,IGE}\right)^2 \qquad (6)$$

  is 0.43 µg/m$^3$ which is well below the limit of 2.00 µg/m$^3$ requested for reference methods for PM mass concentration measurements (EN 16450:2017, 2017).
- (page 14, lines 15-22):
  where LoD is the limit of detection of the TCA08 at a sample flowrate of 16.7 LPM and sample timebase of 1h. In the uncertainty budget of TC measurement with the TCA08 the following sources of uncertainties were not included: (1) Temperature and pressure variations in the sample flow as they are measured by meteorological sensor and included in TC concentration calculations. (2) Temperature and pressure variations in analytical flow as both parameters are measured within NDIR Licor sensor and included in $CO_2$ concentration determination. (3) Sampling artefacts and denuder efficiency: positive/negative artefact phenomenon is recognized by standards EN 12341:2014 and EN 16909:2017, but, as the magnitude of these effects cannot be quantified precisely, they are not considered in the uncertainty budget.

7. **High volume samples were collected for analysis by the two laboratories. Is there information available on the type of the filters used? Was the homogeneity of the samples estimated for this study? Did the laboratories analyze the samples in triplicate or duplicate and were there standard external solutions analyzed like in most of the comparison exercises referenced in the text? Standard solutions may provide an insight if the difference between the TC results of the 2 laboratories was a result of calibration deviations.**

The filters used for offline sample are 150 mm Tissuquartz 2500 Qat-Up produced by Pall Corporation. The homogeneity of samples was tested by ARSO laboratory for 5 random filters from the campaign. In the table (Table 3) below results of the homogeneity test are shown:

| Date | OC [1 st] µg/cm2 | EC [1 st] µg/cm2 | TC [1 st] µg/cm2 | OC [2nd] µg/cm2 | EC [2nd] µg/cm2 | TC [2nd] µg/cm2 | ABS DIFF OC µg/cm2 | ABS DIFF EC µg/cm2 | ABS DIFF TC µg/cm2 |
|---|---|---|---|---|---|---|---|---|---|
| 18/02/2017 | 27.34 | 4.11 | 31.45 | 27.18 | 4.34 | 31.52 | 0.15 | 0.23 | 0.07 |
| 03/03/2017 | 29.15 | 8.22 | 37.37 | 29.77 | 9.96 | 39.73 | 0.62 | 1.74 | 2.37 |
| 08/03/2017 | 17.50 | 7.78 | 25.28 | 17.75 | 7.91 | 25.66 | 0.25 | 0.12 | 0.37 |
| 09/03/2017 | 32.07 | 10.21 | 42.29 | 31.82 | 11.02 | 42.85 | 0.25 | 0.81 | 0.56 |
| 10/03/2017 | 19.78 | 5.82 | 25.60 | 18.51 | 5.69 | 24.20 | 1.27 | 0.13 | 1.40 |

*Table 4 Results of homogeneity test for 5 different filters from the measurement campaign*

Both laboratories use standard calibration solutions and quality control procedures according to standard EN 16909. Additionally, both laboratories were part of ACTRIS exercises and follow ACTRIS guidelines and recommended practices:
- IGE laboratory (Inter-laboratory comparison 2016 and 2017, contact person: Jean Luc Jaffrezo), (ACTRIS, 2016, 2017)

- ARSO laboratory (inter-laboratory comparison 2018, contact person: Judita Burger), (ACTRIS, 2018)

8. **A common practice in comparison exercises is following identical procedures on filter handling, transport and storage for all participants. In the current case filters were first analyzed by the ARSO laboratory and then shipped to IGE for further analysis. Even though the authors mention that "sampling, transport and storage of the filters was done according to the EN 16909:2017" IGE received the filters for analysis at a later period, after additional transport, handling and storage. Could that have contributed to the small uncertainty observed between the reference methods? Wouldn't an approach of dividing the samples and shipping in both labs in parallel have resulted in improved comparability of the two laboratories?**

We agree with the reviewer, it would be better to wait for the end of the campaign, than divide the filters and afterwards analyze them in both laboratories. Nevertheless, all measures were taken to assure that shipped samples to IGE were not contaminated. The measurement campaign was conducted between 7 February and 10 March 2017 at the urban background air quality monitoring station of the Slovenian Environmental Agency (ARSO). After the cartridge with 14 sampled filters was replaced by a cartridge with fresh filters, the sampled filters were stored at ARSO storage facility according to EN 169090:2017. They were analyzed in order by date in three days: 20 Feb 2017, 8 March 2017 and 21 March 2017. Afterwards, approx. 40 mm punches of these filters were stored in sealed petri dishes in shipped to IGE laboratory by express postal service. During shipment the petri dishes containing the filter punches were stored in cooled portable freezer box with temperatures between 5-15 $^{\circ}$C.

9. **Table 3 provides with a wide range of b factors and the recommendation right above is: "the determination needs to be performed for each location and with filters sampled over the time-period of interest". Considering that the b factor is location and season specific, could the authors suggest a typical coverage period range with OC/EC offline analysis in parallel? How often would the b factor have to be re-evaluated per location?**

The proportionality parameter *b* is an effective value with a local and regional component. Usually, the local contribution to concentrations is dominant and the local BC and EC contributions dominate the relationship. The differences in *b* values presented in Table 3 show, that there is a big variation between different rural/regional background sites, and also between the urban sites. This is the reason why similar offline-to-online intercomparison is recommended for every new background site or site with strong mixture of local and regional contribution. The time period of intercomparison should cover seasonal variations in *b* values, for example 2-3 weeks each season. The re-evaluation intercomparison campaign for the certain location should be done if significant changes in the BC emission inventory is expected (traffic or wood burning restrictions, *etc.*). For sites with dominant traffic contribution, where the *b* factor mostly depends on the properties of the vehicle in the fleet, the intercomparison measurements will result in similar *b* values unless a significant fleet change occurs.

Following sentences were added to the manuscript:

- (page 15., lines 11-18)

The proportionality parameter *b* (Eq. 3) is compared with values taken from the literature in Table 3. These values depend on the location, the nature of the aerosol, and the thermal protocol used for analysis. The value of 0.44 which we determined in this study for an urban background site is slightly lower than values for other urban and urban background sites using EUSAAR 2 thermal protocol, and considerably lower than the values for rural sites. The proportionality parameter *b* is an effective value that features a local and a regional contribution of BC and EC. Usually, the local contribution to concentrations is dominant and the local BC and EC contributions dominate the relationship. The differences in *b* values presented in Table 3 show, that there is a big variation between different rural/regional background sites, and also between the urban sites. This is the reason why similar offline-to-online intercomparison is recommended for every new background site or site with strong mixture of local and regional contribution. The time period of the intercomparison should cover seasonal variations in *b* values, for example 2-3 weeks each season. The re-evaluation intercomparison campaign for the certain location should be done if significant changes in the BC emission inventory is expected (traffic or wood burning restrictions, *etc.*) For sites with dominant traffic contribution, where the b factor mostly depends on the properties of the vehicle in the fleet, the intercomparison measurements will result in similar *b* values unless a significant fleet change occurs.

10. **The calculated b factor for this study (0.44) is the lowest among EUSAAR_2 users of the literature listed in Table 3. Further the slope from the OM – eOC comparison of (Figure 11,(b)) is 1.82, and would have been even higher when considering the high negative intercept. Following the ranges provided by the literature in page 17, lines 14-15, these slopes fit better a rural site rather than an urban environment. This comes in contradiction with the characteristics of the selected site which is influenced from traffic emissions. If the estimated b factor was within the literature range that would have further resulted in a lower OM - eOC slope and would fit the literature range better. How would the authors comment on these observed differences compared to the literature?**

There is a big variation between different rural/regional background sites, and also between the urban sites. These differences show that a site and season specific factor *b* needs to be assessed. The obtained OM/OC of 1.82 indicates on mixture of traffic and biomass emissions which agrees with other studies done in Ljubljana (Gjerek et al., 2018; Ogrin et al., 2016). Ljubljana is located in a subalpine basin surrounded by hills. Large forest areas provide a cheap heating source, which represents a government promoted alternative to the use of fossil fuel. While the measurement of black carbon in Ljubljana apportioned to traffic show great variability in concentrations among measurement sites, the contribution of biomass burning is spatially distributed much more homogeneously across wider area in Ljubljana. In winter 2013/14 campaign at 2 urban background measuring sites, study showed averaged value of BC from traffic of 2.5 ± 1.8 µg/m$^3$, while from biomass burning contribution of 1 ± 0.7 µg/m$^3$ (Ogrin et al., 2016).

11. **Following Graph 7, it seems that in the first half of the campaign BC is overestimated while on the second half underestimated, compared to EC. Would there be any interpretations for this observation? Further, around the 5th of March eOC and TC from the TCA08 configuration**

**are overestimated and bBC is underestimated, significantly more from the rest of the data points. Would there be any justification by the authors?**

Detailed analysis of the EC and BC data does not reveal any new interpretation of this observation. Analysis of the 24 h averaged values of the loading compensation parameter k6 and eEC concentrations shown in Figure 2 does not show any significant change of the coating of the BC particles during measurement campaign that could have an effect on the measured eEC concentrations (Drinovec et al., 2017).

[Figure]

*Figure 2 Loading compensation parameter k6 and eEC concentrations during measurement campaign*

Similary, the comparison of the offline analysis for EC on filter by sampling date (Figure 3) or by date of the analysis at the ARSO laboratory (Figure 4) does not show any significant descripancies. At the moment, the temporal variations of descripancy between eEC and offline EC values is unknown.

[Figure]

.

*Figure 3 Comparison of offline EC analysis between ARSO and IGE laboratories sorted by sampling date.*

[Figure]

*Figure 4 Comparison of offline EC analysis between ARSO and IGE laboratories sorted by the date of analysis by ARSO laboratory.*

12. **Page 15, lines 19-21, suggest that the intercept due to the eOC positive artefact can be neglected based on the comparability with the OC intercept of the two independent laboratory measurements. Nevertheless, as discussed earlier in the text, the eOC intercept is systematic and attributed to the denuder performance. It should also be noted that the eOC concentrations are compared to the average values between the two laboratories. It would be more appropriate if the artefacts observed for a new instrument would not be overlooked but rather investigated further and be dealt with. The use of two candidate method analyzers and an extended data set would be required for in-depth analysis. The above comes also in contradiction with the conclusion of page 20, line 12: "the correlation analysis showed very high agreement between eOC and eEC to the EC and OC". It seems that there is room for improvement in agreement once the artefact issues are resolved.**

We agree with the reviewer. We are aware of the importance of OC positive/negative artefact when comparing online OC/EC instruments to analysis of the 24h filters. In our current study (Gregorič et al., 2020), positive and negative artefact on quartz filters depending on sample time base, face velocity, number of denuders, chemical deposition of aerosol is investigated in details in order to quantify the magnitude of these effects. In this publication guidelines and recommendations on how to asses positive/negative artefact with TCA08 and how to use this knowledge for more quality measurement data will be described. Again, positive artefact and denuder efficiency is recognized by standards EN 12341:2014 and EN 16909:2017: but not included in the uncertainty budget.

Please refer to points 4, 5, and 6 for details.

13. **Technical corrections:**
    **Page 2, line 26: tTe.**
    - The amount of OC converted into PC during the analysis depends on many factors, including the amount and type of organic compounds, the sources of air pollution,

[revised manuscript text omitted]

---

## Author Comment (AC2) · 27 Feb 2020

**Point-to-point response to Referee #2 (RC2)**

We are grateful for yours comments on how to strengthen our manuscript. Please find attached revised manuscript and see our point-to-point response to your comments below:

1. **Introduction: The intro part mainly discussed the definitions of different carbon fractions and various protocols for measuring the carbon fractions. However, some of the content is repetitive (e.g. it was discussed in Page 2 Line 26 to Page 3 Line 2 that a few factors will influence the OC-EC split while similar points were mentioned again in Page 3 Lines 9–27). The authors also listed out three protocols that have been widely used in different regions of the world (Page 2 Lines 16–24). Since in the following sections the online data were compared with the offline data obtained from the EUSAAR2 protocol, readers might expect to see more discussions on the specific differences among the protocols (EUSAAR2 vs. IMPROVE_A, EUSAAR2 vs. NIOSH).**

   We believe that discussion on the specific difference among protocols is beyond the scope of this paper and was very well described in Karanasiou et al. (2015). We focus on operational definition aspects and harmonization between different experimental approaches.

   Following sentence was added to the manuscript (page 2, lines 24-25):

   - This protocol has recently became part of the European standard for the determination of OC-EC in PM2.5 samples (EN 16909:2017, 2017). Detailed discussion on the specific difference among protocols can be found elsewhere (Cavalli et al., 2010; Karanasiou et al., 2015).

2. **Section 2.2: How is the performance of external calibration (using sucrose or KHP or other chemicals) by TCA08? What is the maximum carbon concentration tested?**

   External calibration of TCA08 was performed with punches of ambient filters with known TC content. This is done to simplify the calibration procedure on the field and to achieve better calibration accuracy (ambient filter includes mixture of different EC and OC fractions). Carbon calibration constant is defined with the slope between measured integrated pulse of $CO_2$ by TCA08 and known TC mass content of the filter punch (Figure 1). The maximum carbon content tested in the TCA08 was up to 100 µg of TC.

[Figure]

*Figure 1 External calibration of TCA08 - determination of carbon calibration constant*

3. **Section 3.3: What is the "carbon calibration factor"? How is it derived? What's the loading effect compensation algorithm used for treating AE33 data in this study? Will different algorithms introduce uncertainties?**

Carbon calibration factor mentioned in paragraph in section 3.3 is carbon calibration constant $C_{carb}$ from Eq. 4. It is defined as the slope between TC mass on the punch of ambient filter with known TC content and the integrated value of the $CO_2$ signal measured by the NDIR detector (Figure 1).

Carbon calibration factor was changed to carbon calibration constant (page 14, line 5):
- The uncertainty $u_{TCA}$ associated with the TC data from the TCA08 includes individual uncertainty sources of the carbon calibration constant $C_{carb}$; the uncertainty of the analytic flow measurement; and the uncertainty of the signal and blank $CO_2$ peak measurement (Eq. 4).

Loading effect compensation algorithm used for treating AE33 data in this study is an real time "dual spot" loading compensation (Drinovec et al., 2015, 2017). The inlet air stream of the AE33 is split, and the sample is collected on two filter spots concurrently. The flow through each of the two spots is different, so the loading rates on the respective sample spots are different. Different loading rates cause the accumulation of the sample to be different between the two spots, resulting in a different magnitude of filter loading effect (FLE) between the spots. Measurement of FLE enables the compensation of the data – using the parametrization described in Drinovec et al. (2015), the compensation parameter $k$ can be derived. We believe this method is the most appropriate one as it measures loading effect with the same time resolution as measuring black carbon concentrations and it does not make any assumptions. Using different compensation algorithms will introduce uncertainties based on the assumptions they use.

A following sentence was added to the manuscript (page 14, line 26):

- Figure 7 shows the regression of the off-line thermo-optical analysis of samples for EC (from the ARSO and IGE laboratories, using the EUSAAR_2 protocol) with the 24-hour averaged BC (Aethalometer data) obtained during the field campaign period. An AE33 integrated "dual spot" real-time loading compensation algorithm was used for BC data treatment (Drinovec et al., 2015).

4. **Section 3.4: It can be seen from this work and from previous literature data (Table 3) that the relationship between BC and EC is location dependent. If the aerosol composition at a certain location has a very clear temporal variation pattern, the parameter b could be sensitive to the sampling time period as well. Since the PM monitoring networks usually adopt the filter-based sampling approach followed by offline laboratory analysis and the historical dataset was very likely obtained from offline measurement, do the authors suggest that every time the online TC-BC system is deployed to one sampling location, the online-offline comparison needs to be conducted to derive the b value so that the measured data can be compared to other dataset?**

We agree with the reviewer. The sampling time period for offline filters can affect the b factor as well. We followed a common practice for sampling period; sampling start time was at 00.00 am and sampling stop time was at23.55 pm each day. During 5-minute idle period, the sampler automatically stored sampled filter and replaced it with the new one.

[revised manuscript text omitted]

---

## Referee Report (RR1)

**Amt-2019-376 review anonymous author**

Many thanks to the authors for the consideration of the comments provided and for delivering the revised manuscript. I would like to kindly bring their attention to some of the suggestions that where partially handled and require additional clarifications or elaboration. Namely:

1) As mentioned in the abstract of the paper "*The concentration of particulate organic carbon (OC) is determined by subtracting black carbon concentration, concurrently measured optically by an Aethalometer®, from the total carbon concentration measured by the TCA08*". This is also described in detail in the first two paragraphs of the method and instrument description section (section 2.1). The method described, as also mentioned by the authors at the third paragraph of section 2.1, has been earlier introduced in Bauer et al. (2009) and elsewhere, e.g. Arhami et al. (2006). No doubt that the current method application and the new instrumentation described in the paper introduces novelties and convenience for the user as listed in detail by the authors. Nevertheless, I would advise modifying the title and related text from "new TC-BC method" to either "new application of the TC-BC method" or "new instrument introduction for the TC-BC method" or similar.

2) In line 24 of the abstract as well as in section 3 it is mentioned that the *equivalence* of the newly introduced instrument and off-line thermo-optical OCEC reference method has been evaluated. The authors attempted to follow the EN16450:2017 but as mentioned in the text the norm procedure was not properly applied. Since an equivalence procedure can return two outcomes, pass or fail, it would be fair for the reader to clearly state which of the two was the case for this attempt. I suggest to either modify the text accordingly in order to include this information or remove the equivalence term.

3) In deviation from EN16450:2017 only one candidate instrument application was included in the campaign. The authors mention in-house tests as an alternative. This is not a valid argument for a number of reasons: non-compliant with EN16450:2017, different aerosol type sampled, different conditions between in-house and field campaign, to name a few. The in-house tests and data obtained are of course welcome and can be presented in detail but in any case, cannot serve as an alternative. The text should be modified accordingly.

4) The uncertainty limit value of 2.00 $\mu g/m^3$ mentioned in page 11 line 12 originates from calculations of PM reference methods where limit values of 30 $\mu g/m^3$ or more apply. This value is not directly applicable for TC measurements which is a fraction of PM. Additionally one would expect a more stringent limit value for ambient TC concentrations compared to PM. Both the above will result in a significantly lower uncertainty limit value in the future. There is clearly the need of introducing a method specific uncertainty limit value but unfortunately this is not there yet. It would be best if the above are clarified more in the text. The uncertainty limit value of 2.00 $\mu g/m^3$ should serve as an indication and not as a direct criterion for compliance. Finally, the calculation of the method uncertainty of 0.43 $\mu g/m^3$ is not valid since less than 40 data points were used for its calculation, which does not meet the requirements of EN16450:2017.

5) It is common practice in comparison exercises, e.g. as organized by ACTRIS, for laboratories to receive simultaneously parts of the filter and perform triplicate analysis. Further, standard solutions are also included in order to investigate the analyzer calibration of each participant. It seems that in the comparison organized by the authors, the first laboratory performed single

analysis and then forwarded the whole filter to the second laboratory for another single analysis. Further no standard solution was included. EN16909:2017 includes the following in 7.2 NOTE: *"OC concentration may change depending on handling…"*. The above might have added to the observed differences between the results of the two laboratories and itwould provide the whole picture to the reader if they were mentioned clearly in the text.

6) Section 3.3, last sentence, item 3,mentions that "*Sampling artefacts and denuder efficiency: positive/negative artefact phenomenon is recognized by standards EN 12341:2014 and EN 16909:2017, but, as the magnitude of these effects cannot be quantified precisely, they are not considered in the uncertainty budget."* Indeed, that is the case for low volume samplers without the application of a denuder for a 24-hour sampling period. As stated elsewhere in the manuscript, e.g. page 13 last paragraph, the TCA method faces a pronounced VOC adsorption positive artifact due to reduced sampling times. Considering the above, a more appropriate budget uncertainty calculation would include the uncertainty due to the positive artifact which in any case is not negligible.

7) In section 3.5 an offset of 1.33 $\mu g/m^3$ is found for the comparison between eOC and OC. When considering that the average measurements of the laboratories where applied as the reference method this will probably result at a +/- 0.4 $\mu g/m^3$ offsets for each laboratory from the their average results, which as mentioned earlier might originate from handling variation of filters or analyzer calibration deviation. The observed offset of 1.33 $\mu g/m^3$ is more than 3 times greater than 0.4 $\mu g/m^3$. Further when comparing the Pearson correlations for OC measurements including and excluding the intercept this remains 0.99 for the laboratories while it drops from 0.98 to 0.94 for the eOC to OC comparison. Finally, there is a clear systematic bias which the authors attribute to a dominant positive artefact related to VOCs and denuder efficiency. Despite the above clear indications of a significant limitation of the application the authors conclude that the observed offset can be neglected. I recommend revision of the current section accordingly. For your consideration EN16450:2017 includes a section where describes the calculations for determining the requirement of an intercept correction or not and also lists the following criterion: Calibration needs not to be performed when the value of the intercept is 1,0 $\mu g/m^3$ ≤ a ≤ 1,0 $\mu g/m^3$

8) Correction in references: ARSO laboratory participated in 2016-2 ACTRIS comparison exercise and not in 2018-1.

References

Jace J. Bauer, Xiao-Ying Yu, Robert Cary, Nels Laulainen & Carl Berkowitz (2009) Characterization of the Sunset Semi-Continuous Carbon Aerosol Analyzer, Journal of the Air & Waste Management Association, 59:7, 826-833, DOI: 10.3155/1047-3289.59.7.826

Mohammad Arhami, Thomas Kuhn, Philip M. Fine, Ralph J. Delfino and Constantinos Sioutas (2006) Effects of Sampling Artifacts and Operating Parameters on the Performance of a Semicontinuous Particulate Elemental Carbon/Organic Carbon Monitor, Environ. Sci. Technol. 2006, 40, 3, 945-954

---

## Author Response (AR2)

Dear reviewer,

We are grateful for your comments on how to strengthen our manuscript. The goal of this paper is to introduce and evaluate a new instrument, which can be used for the TC-BC method. We did not try to show the equivalence of the TC-BC method exactly following the standard EN16450 (EN 16450:2017, 2017), but the tools and methods developed in this standard (intended for PM10 and/or PM2.5 measurements, and not OC/EC or CM) are useful to show the performance of the TC-BC method and evaluation of the respective instruments. We changed the text in the manuscript accordingly: "new TC-BC" method was changed to "TC-BC method", term "equivalence" was removed from the manuscript. Please find our point to point response below.

1) **As mentioned in the abstract of the paper "The concentration of particulate organic carbon (OC) is determined by subtracting black carbon concentration, concurrently measured optically by an Aethalometer®, from the total carbon concentration measured by the TCA08". This is also described in detail in the first two paragraphs of the method and instrument description section (section 2.1). The method described, as also mentioned by the authors at the third paragraph of section 2.1, has been earlier introduced in Bauer et al. (2009) and elsewhere, e.g. Arhami et al. (2006). No doubt that the current method application and the new instrumentation described in the paper introduces novelties and convenience for the user as listed in detail by the authors. Nevertheless, I would advise modifying the title and related text from "new TC-BC method" to either "new application of the TC-BC method" or "new instrument introduction for the TC-BC method" or similar.**

   - The title of the manuscript was changed accordingly.

   ## The new instrument using TC-BC method for the online measurement of carbonaceous aerosols

   - p. 1, lines 16-17: **Abstract.** We present the newly developed Total Carbon Analyzer (TCA08), and a  method for online speciation of carbonaceous aerosol with a high time resolution.
   - p. 3, lines 33-34: The  TC-BC method presented in this study is an easy-to-deploy and low maintenance continuous measurement technique for the high time resolution determination of organic and elemental carbon in different PM fractions (PM10, PM2.5 and PM1).
   - p.4 lines 5-6: In this study we present the newly developed application of TC-BC method, which combines an optical method for measuring mass equivalent black carbon (eBC) by the AE33 Aethalometer (Drinovec et al., 2015; Hansen et al., 1984), and a thermal method for total carbon (TC) determination by a new instrument, the Total Carbon Analyzer TCA08, developed and commercialized by Aerosol d.o.o. (Ljubljana, Slovenia).
   - p. 4, lines 19-24: Although one can find conceptual similarities between method presented in Bauer et al., 2009 (and references therein) and TC- BC method presented in this study, the new application of the method takes the advantage of decoupling thermal and optical method into two separate instruments, both dedicated for different measurements. With this, the  TC-BC method has higher time resolution, no sampling dead time, online loading nonlinearity compensation for eBC measurements (Drinovec et

al., 2017) and is more convenient for field measurements as the thermal measurement is done without fragile quartz cross oven, high purity gases and catalyst.

2) **In line 24 of the abstract as well as in section 3 it is mentioned that the equivalence of the newly introduced instrument and off-line thermo-optical OCEC reference method has been evaluated. The authors attempted to follow the EN16450:2017 but as mentioned in the text the norm procedure was not properly applied. Since an equivalence procedure can return two outcomes, pass or fail, it would be fair for the reader to clearly state which of the two was the case for this attempt. I suggest to either modify the text accordingly in order to include this information or remove the equivalence term.**

We did not try to show the equivalence of the TC-BC method exactly following EN 16450 (EN 16450:2017, 2017), but the tools and methods developed in this standard are useful to show the performance of the TC-BC method and evaluate the instruments as compared to the reference method. Terminologically, we think that "the equivalence … has been evaluated" is an appropriate description. However, it is evident from the reviewer's comments that this terminology can be misunderstood, and we have removed the term "Equivalence" from the manuscript.

- p. 1, lines 24-26: The performance of this online method relative to the standardized off-line thermo-optical OC-EC method and respective instruments was evaluated during a winter field campaign at an urban background location in Ljubljana, Slovenia.
- p. 3-4, lines 43, 1-2: The performance of this online method relative to the standardized off-line thermo-optical OC-EC method and respective instruments is evaluated through analysis of regression models of the various compared methods.

3) **In deviation from EN16450:2017 only one candidate instrument application was included in the campaign. The authors mention in-house tests as an alternative. This is not a valid argument for a number of reasons: non-compliant with EN16450:2017, different aerosol type sampled, different conditions between in-house and field campaign, to name a few. The in-house tests and data obtained are of course welcome and can be presented in detail but in any case, cannot serve as an alternative. The text should be modified accordingly.**

Term "equivalence" is removed from the manuscript (See point 2). In our previous revision of the manuscript we explained that we did not follow EN16450:2017, which is intended for PM10/PM2.5 measurements, but we used the tools and methods developed in the standard. A supplement was added.

- p. 9, lines 26-30: As there is no standard for reference method for online measurement of OC and EC concentrations available at the time of the writing of this manuscript, we  used tools and methods developed in EN16450:2017 and choose EN 16909:2017 as the reference method. Nevertheless, a proper application of EN16450:2017 would require a minimum of 40 valid data pairs with the further requirement of two candidate applications for each type testing application. Additionally, the same standard further

describes requirements related to the number of locations and the concentration range of data points.

- p. 9, lines 33-36: Furthermore, we used only one set of instrument for the candidate method comparison. Both instruments, TCA08 and AE33, are compared to the reference set of instruments after their assembly as one of the tests during final inspection procedure (in-house defined requirements for successful intercomparison between new and reference set of instruments are: 1. TCA08: TC concentration range up to 75.000 ng/m$^3$, slope between 0.95-1.05, $R^2$ above 0.98 ; 2. AE33: eBC concentrations up to 25.000 ng/m$^3$; slope between 0.95-1.05, $R^2$ above 0.98, Table S1).

- Table S1 in the Supplement was added:

| | Instruments serial number | Slope | $R^2$ | N | eBC$_{max}$ (ng/m$^3$) | eBC$_{min}$ (ng/m$^3$) |
|---|---|---|---|---|---|---|
| 1 | AE33-S08-01036 | 1.01 | 1.00 | 6368 | 19666 | 265 |
| 2 | AE33-S08-01037 | 1.00 | 1.00 | 6368 | 19478 | 220 |
| 3 | AE33-S08-01038 | 1.02 | 1.00 | 6368 | 21988 | 139 |
| 4 | AE33-S08-01039 | 1.02 | 1.00 | 6368 | 20126 | 198 |
| 5 | AE33-S08-01040 | 1.00 | 1.00 | 6368 | 21338 | 224 |
| 6 | AE33-S08-01041 | 0.99 | 1.00 | 6368 | 20384 | 231 |
| 7 | AE33-S08-01042 | 1.04 | 1.00 | 6320 | 22141 | 123 |
| 8 | AE33-S08-01043 | 1.04 | 1.00 | 6368 | 21380 | 286 |
| 9 | AE33-S08-01044 | 1.03 | 1.00 | 6368 | 21249 | 223 |
| 10 | AE33-S08-01045 | 1.02 | 1.00 | 6368 | 21050 | 172 |
| 11 | AE33-S08-01046 | 0.99 | 1.00 | 6368 | 20433 | 267 |
| 12 | AE33-S08-01047 | 1.03 | 1.00 | 6368 | 20583 | 230 |
| 13 | AE33-S08-01048 | 1.03 | 1.00 | 6368 | 21061 | 237 |
| 14 | AE33-S08-01049 | 1.03 | 1.00 | 6320 | 20699 | 249 |
| 15 | AE33-S08-01050 | 1.02 | 1.00 | 6320 | 20064 | 243 |

| | Instruments serial number | Slope | $R^2$ | N | TC$_{max}$ (ng/m$^3$) | TC$_{min}$ (ng/m3) |
|---|---|---|---|---|---|---|
| 1 | TCA08-S00-0131 | 1.02 | 1.00 | 65 | 45856 | 2732 |
| 2 | TCA08-S00-0132 | 1.01 | 1.00 | 65 | 45803 | 2732 |
| 3 | TCA08-S00-0133 | 1.00 | 1.00 | 19 | 67274 | 6286 |

| | | | | | | |
|---|---|---|---|---|---|---|
| 4 | TCA08-S00-0134 | 1.00 | 1.00 | 19 | 67274 | 6286 |
| 5 | TCA08-S00-0135 | 1.00 | 1.00 | 19 | 67274 | 6286 |
| 6 | TCA08-S00-0136 | 0.99 | 1.00 | 139 | 76684 | 2498 |
| 7 | TCA08-S00-0137 | 1.00 | 1.00 | 65 | 51119 | 2592 |
| 8 | TCA08-S00-0138 | 0.99 | 1.00 | 65 | 51119 | 2592 |
| 9 | TCA08-S00-0139 | 0.96 | 1.00 | 65 | 51119 | 2592 |
| 10 | TCA08-S00-0140 | 1.02 | 1.00 | 62 | 73190 | 2644 |
| 11 | TCA08-S00-0141 | 1.00 | 1.00 | 22 | 64579 | 3336 |
| 12 | TCA08-S00-0142 | 1.01 | 1.00 | 136 | 76466 | 2823 |
| 13 | TCA08-S00-0143 | 1.00 | 1.00 | 22 | 64579 | 3336 |
| 14 | TCA08-S00-0144 | 1.00 | 1.00 | 22 | 64609 | 3336 |
| 15 | TCA08-S00-0145 | 0.97 | 1.00 | 137 | 72992 | 2823 |

Table S1: Example of intercomparison results of TCA08 and AE33 after their assembly to the reference set of instruments as one of the tests during final inspection procedure. In-house defined requirements for successful intercomparison between new and reference set of instruments are: 1. TCA08: TC concentration range up to 75.000 ng/m3, slope between 0.95-1.05, $R^2$ above 0.98 ; 2. AE33: eBC concentrations up to 25.000 ng/m3; slope between 0.95-1.05, $R^2$ above 0.98.

4) **The uncertainty limit value of 2.00 μg/m3 mentioned in page 11 line 12 originates from calculations of PM reference methods where limit values of 30 μg/m3 or more apply. This value is not directly applicable for TC measurements which is a fraction of PM. Additionally one would expect a more stringent limit value for ambient TC concentrations compared to PM. Both the above will result in a significantly lower uncertainty limit value in the future. There is clearly the need of introducing a method specific uncertainty limit value but unfortunately this is not there yet. It would be best if the above are clarified more in the text. The uncertainty limit value of 2.00 μg/m3 should serve as an indication and not as a direct criterion for compliance. Finally, the calculation of the method uncertainty of 0.43 μg/m3 is not valid since less than 40 data points were used for its calculation, which does not meet the requirements of EN16450:2017.**

In the previous revision we added a paragraph, where we describe discrepancies between our approach and an application of EN 16450:2017 to show equivalence. (See points 2 and 3). Additionally, we clarify the in text, that PM uncertainty limit can be used for indication only.

- The uncertainty $u_{RM}$ between the reference methods for TC

$$u_{RM}^2 = \frac{1}{2n}\sum_{i=1}^{n}\left(TC_{i,ARSO} - TC_{i,IGE}\right)^2 \tag{6}$$

is 0.43 μg/m³ which is well below the limit of 2.00 μg/m³ requested for reference methods for PM mass concentration measurements (EN 16450:2017, 2017). As there is no method

specific uncertainty limit for TC available yet, the limit for PM can serve as an indication only, not for direct criterion of compliance.

5) **It is common practice in comparison exercises, e.g. as organized by ACTRIS, for laboratories to receive simultaneously parts of the filter and perform triplicate analysis. Further, standard solutions are also included in order to investigate the analyzer calibration of each participant. It seems that in the comparison organized by the authors, the first laboratory performed single analysis and then forwarded the whole filter to the second laboratory for another single analysis. Further no standard solution was included. EN16909:2017 includes the following in 7.2 NOTE:** *"OC concentration may change depending on handling…"*. **The above might have added to the observed differences between the results of the two laboratories and it would provide the whole picture to the reader if they were mentioned clearly in the text.**

The DIGITEL sampler maintenance and sample storage were handled by ARSO (Slovenian Environmental Agency) technician staff, as it was situated at their premises, where the analysis was also performed, so no shipping was necessary – this is similar to the analysis performed at JRC Ispra with samples from their site. This is also the reason why samples were first analyzed at ARSO and afterwards at IGE laboratory. Nevertheless, as both laboratories are regularly involved in the ACTRIS actions, they are familiar with the latest guidelines and instructions on the sample handling and storage, sampler and OC-EC instrument maintenance and calibration. Both laboratories preform daily validation sucrose tests. Sucrose validations showed measurements within 5% for the days our samples were measured at both laboratories, hence no recalibration was needed. The ARSO laboratory also preformed five duplicate measurements of the punches from the same filters, all results were within 5 % (See Table 4, 'Point-to-point response to Referee #1 (RC1)', Martin Rigler, 27 Feb 2020).

Text in the manuscript was changed accordingly:

- p. 11-12, lines 16-8:  However, the difference in slope for OC and consequently for TC is around 10%, with a negative intercept value of around -0.80 μg/m$^3$ for OC and TC (using linear orthogonal regression model with intercept) which can indicate possible differences in instrument calibration, suboptimal performance of one of the instruments (featuring artefacts) or inadequate filter sample handling.  The EN 16909:2017 standard includes in chapter 7.2 a note that OC concentration may change depending on the sample handling. Both laboratories preform daily calibration constant validation with sucrose solution. Sucrose validations showed values within 5% of theoretical carbon content in the sucrose solution for the days these samples were analyzed at both laboratories. Hence, no calibration was needed and performed before filters from this study were analyzed.  The ARSO laboratory also preformed five duplicate measurements of the punches from the same filters, all results were within 5 %. The filter samples were first measured in ARSO laboratory, and then shipped to IGE laboratory. Sampling, transport and storage of the filters were done according to EN 16909:2017 (EN 16909:2017, 2017).

6) **Section 3.3, last sentence, item 3, mentions that "Sampling artefacts and denuder efficiency: positive/negative artefact phenomenon is recognized by standards EN 12341:2014 and EN 16909:2017, but, as the magnitude of these effects cannot be quantified precisely, they are not considered in the uncertainty budget." Indeed, that is the case for low volume samplers without the application of a denuder for a 24-hour sampling period. As stated elsewhere in the manuscript, e.g. page 13 last paragraph, the TCA method faces a pronounced VOC**

**adsorption positive artifact due to reduced sampling times. Considering the above, a more appropriate budget uncertainty calculation would include the uncertainty due to the positive artifact which in any case is not negligible.**

In paragraph "**3.3 TCA08 method uncertainty**", uncertainty of the TCA08 instrument is discussed. The concentration of organic gases is a property of the sampled air and can have significant temporal variations. Additionally, offline filters samples (high volume and low volume) and consequent OC/EC analysis have the same problem, which is not included in their uncertainty balance. There is no actual reference method for the sampling artefact. Applying denuders in the 24h sampling system for offline OC-EC analysis may increase the magnitude of negative volatilization artifacts, since lowered organic vapor pressures favor volatilization of organic carbon from particles already collected on the filter (Arhami et al., 2006). Therefore, uncertainty of sampling artefact for one TCA measurement cannot be determined, but can be estimated by denuder efficiency test or method described by Arhami et al, 2006. In our current study (Gregorič et al., 2019, 2020), positive and negative artefact on quartz filters depending on sample time base, face velocity, number of denuders, chemical deposition of aerosol is investigated in great detail. We will also provide estimation of contribution of positive artefact to uncertainty budget as a function of sampling time base and chemical composition of organic gases and aerosol.

Following text was added to the manuscript:

- p. 15, lines 3-6: (3) Sampling artefacts and denuder efficiency: positive/negative artefacts phenomenon are recognized by standards EN 12341:2014 and EN 16909:2017, but as the magnitude of these effects cannot be quantified precisely, they are not considered in the uncertainty budget. However, by using the denuder efficiency routine described in chapter 2.3 and Eq. 5, one can estimate the absolute value of positive artefact and set the sampling time base accordingly to reduce contribution of this phenomenon to the uncertainty budget. Furthermore, introducing an inline Teflon filter at the sample inlet of one of the chambers, provides semi-continuous measurement (every second measurement) of positive artefact. The details of this method are described in Arhami et al., 2006. For this method, the denuder is installed in the common flow stream for both channels, while the inline Teflon filter is positioned only in the flow stream passing to Channel 1 (Fig. S1). Example of evaluation of denuder breakthrough contribution to the TC measurement uncertainty with inline Teflon filter method is shown in Figure S2.

Supplement was added:

2. **TCA08 setup for semi-continuous denuder breakthrough determination.**

[Figure]

**Figure S1: TCA08 setup when (a) sampling and (b) performing semi-continuous denuder breakthrough measurement. Note that the tubing length is identical in both setups. This permits the test to be performed at a permanent installation without disturbing the inlet plumbing.**

**3.** **TCA08 setup for semi-continuous denuder breakthrough determination.**

[Figure]

**Figure S2: Example of evaluation of denuder breakthrough contribution to the TC measurement uncertainty with inline Teflon filter method (see Fig. S1 (b) for TCA08 setup). TC$_{DB}$ is measured in chamber 1 where sample air stream passes denuder and inline Teflon filter. The TCA08 was operated on a 1-hour time-base, sampling PM2.5 fraction at 16.7 LPM. The measurement campaign was conducted between 18 December 2019 and 4 January 2020 at the urban background air quality monitoring of Aerosol d.o.o. company at 46.0715°N, 14.5018°E, elevation 302 m.**

7) **In section 3.5 an offset of 1.33 µg/m3 is found for the comparison between eOC and OC. When considering that the average measurements of the laboratories where applied as the reference method this will probably result at a +/- 0.4 µg/m3 offsets for each laboratory from the their average results, which as mentioned earlier might originate from handling variation of filters or analyzer calibration deviation. The observed offset of 1.33 µg/m3 is more than 3 times greater than 0.4 µg/m3. Further when comparing the Pearson correlations for OC measurements including and excluding the intercept this remains 0.99 for the laboratories while it drops from 0.98 to 0.94 for the eOC to OC comparison. Finally, there is a clear systematic bias which the authors attribute to a dominant positive artefact related to VOCs and denuder efficiency. Despite the above clear indications of a significant limitation of the application the authors conclude that the observed offset can be neglected. I recommend revision of the current section accordingly. For your consideration EN16450:2017 includes a section where describes the calculations for determining the requirement of an intercept correction or not and also lists the following criterion: Calibration needs not to be performed when the value of the intercept is 1,0 µg/m3 ≤ a ≤ 1,0 µg/m3**

As the reviewer has mentioned in his comment no. 4, one cannot apply the PM uncertainty limit values for TC measurements directly, and we can use it only for indication. For similar reasons, the intercept calibration criteria for PM cannot be used for TC and OC re-calibration. We understand the reviewer that the offset cannot be neglected and agree that it must be examined in detail. For future TCA08 and AE33 measurement campaigns with expected daily concentrations of OC and TC below 5 µg/m$^3$ we recommend longer denuder efficiency tests or a test with inline Teflon filter (Arhami et al., 2006) to estimate the contribution of positive artefact and determine the appropriate sample time base.

Text in the manuscript was changed accordingly:

- p. 17, lines 9-27: Online eOC measurements can be derived using the above EC-BC correlation plot to assign the appropriate operational value of the parameter *b*; the online BC data; and the online TCA data. Figure 8 shows the correlation between online OC$_{TC-BC}$ and offline OC derived from the 24-hour filter samples analyzed with a thermo-optical OC-EC analyzer. These results show that when using an appropriate value of *b*, the TC – BC Method yields online data for the eOC content of ambient aerosols that agree very well with conventional offline thermal analyses. The offset *i* = 1.33 ± 0.18 µg/m$^3$ lies in the same range as that determined by TC correlation analysis, which confirms that organic carbon is the origin of the offset in the correlation plots in Figs. 6 and 8. The offset is also comparable to that determined by the inter-laboratory comparison of off-line filter analyses (offset OC$_{ARSO}$-OC$_{IGE}$: $i_1$ = -0.81 ± 0.12 µg/m$^3$, offset eOC-OC $i_2$ = 1.33 ± 0.18 µg/m$^3$). The in-depth analysis of the relative difference between OC from 24 h filters and eOC determined by online measurement as TC-*b*BC shown in Fig. 9 reveals that the positive

artefact can be the dominant apparent source of OC for days with very low OC concentrations (< 5 µg/m$^3$) in comparison to offline 24 h filters, for which also negative artefact (desorption of VOCs) can occur. This leads to the importance of regular denuder efficiency/breakthrough determination (Figs. 3, S1 and S2), and consequent appropriate sample time base set-up, according to OC concentration and denuder breakthrough value. For this campaign, a longer sample time base and/or usage of two denuder monoliths in TCA08 would decrease the offset and reduce its contribution to the overall uncertainty budget of eOC measurement. For 11 of the 31 days (OC < 5 µg/m$^3$) in this campaign, 2 h sample time base should be used. As we found out in this study, for field campaigns with daily TC or OC concentrations below 5 µg/m$^3$, it is strongly recommended to preform longer denuder efficiency tests or test with inline Teflon filter (Arhami et al., 2006) to estimate the contribution of positive artefact and determine appropriate sample time base.

8) **Correction in references: ARSO laboratory participated in 2016-2 ACTRIS comparison exercise and not in 2018-1.**

References were changed accordingly:
- p. 24, lines 1-2:

[revised manuscript text omitted]

**1. Example of in-house intercomparison results of TCA08 and AE33 instruments after their assembly to the reference set of instruments**

|  | TCA08 serial number | Slope | $R^2$ | N | $TC_{max}$ (ng/m$^3$) | $TC_{min}$ (ng/m3) |
|---|---|---|---|---|---|---|
| 1 | TCA08-S00-0131 | 1.02 | 1.00 | 65 | 45856 | 2732 |
| 2 | TCA08-S00-0132 | 1.01 | 1.00 | 65 | 45803 | 2732 |
| 3 | TCA08-S00-0133 | 1.00 | 1.00 | 19 | 67274 | 6286 |
| 4 | TCA08-S00-0134 | 1.00 | 1.00 | 19 | 67274 | 6286 |
| 5 | TCA08-S00-0135 | 1.00 | 1.00 | 19 | 67274 | 6286 |
| 6 | TCA08-S00-0136 | 0.99 | 1.00 | 139 | 76684 | 2498 |
| 7 | TCA08-S00-0137 | 1.00 | 1.00 | 65 | 51119 | 2592 |
| 8 | TCA08-S00-0138 | 0.99 | 1.00 | 65 | 51119 | 2592 |
| 9 | TCA08-S00-0139 | 0.96 | 1.00 | 65 | 51119 | 2592 |
| 10 | TCA08-S00-0140 | 1.02 | 1.00 | 62 | 73190 | 2644 |
| 11 | TCA08-S00-0141 | 1.00 | 1.00 | 22 | 64579 | 3336 |
| 12 | TCA08-S00-0142 | 1.01 | 1.00 | 136 | 76466 | 2823 |
| 13 | TCA08-S00-0143 | 1.00 | 1.00 | 22 | 64579 | 3336 |
| 14 | TCA08-S00-0144 | 1.00 | 1.00 | 22 | 64609 | 3336 |
| 15 | TCA08-S00-0145 | 0.97 | 1.00 | 137 | 72992 | 2823 |

| | AE33 serial number | Slope | $R^2$ | N | eBC$_{max}$ (ng/m$^3$) | eBC$_{min}$ (ng/m3) |
|---|---|---|---|---|---|---|
| 1 | AE33-S08-01036 | 1.01 | 1.00 | 6368 | 19666 | 265 |
| 2 | AE33-S08-01037 | 1.00 | 1.00 | 6368 | 19478 | 220 |
| 3 | AE33-S08-01038 | 1.02 | 1.00 | 6368 | 21988 | 139 |
| 4 | AE33-S08-01039 | 1.02 | 1.00 | 6368 | 20126 | 198 |
| 5 | AE33-S08-01040 | 1.00 | 1.00 | 6368 | 21338 | 224 |
| 6 | AE33-S08-01041 | 0.99 | 1.00 | 6368 | 20384 | 231 |
| 7 | AE33-S08-01042 | 1.04 | 1.00 | 6320 | 22141 | 123 |
| 8 | AE33-S08-01043 | 1.04 | 1.00 | 6368 | 21380 | 286 |
| 9 | AE33-S08-01044 | 1.03 | 1.00 | 6368 | 21249 | 223 |
| 10 | AE33-S08-01045 | 1.02 | 1.00 | 6368 | 21050 | 172 |
| 11 | AE33-S08-01046 | 0.99 | 1.00 | 6368 | 20433 | 267 |
| 12 | AE33-S08-01047 | 1.03 | 1.00 | 6368 | 20583 | 230 |
| 13 | AE33-S08-01048 | 1.03 | 1.00 | 6368 | 21061 | 237 |
| 14 | AE33-S08-01049 | 1.03 | 1.00 | 6320 | 20699 | 249 |
| 15 | AE33-S08-01050 | 1.02 | 1.00 | 6320 | 20064 | 243 |

**Table 1: Example of intercomparison results of TCA08 and AE33 after their assembly to the reference set of instruments as one of the tests during final inspection procedure. In-house defined requirements for successful intercomparison between new and reference set of instruments are: 1. TCA08: TC concentration range up to 75.000 ng/m$^3$, slope between 0.95-1.05, $R^2$ above 0.98 ; 2. AE33: eBC concentrations up to 25.000 ng/m3; slope between 0.95-1.05, $R^2$ above 0.98.**

**2. TCA08 setup for semi-continuous denuder breakthrough determination.**

[Figure]

**Figure S1: TCA08 setup when (a) sampling and (b) performing semi-continuous denuder breakthrough measurement. Note that the tubing length is identical in both setups. This permits the test to be performed at a permanent installation without disturbing the inlet plumbing.**

**3. TCA08 setup for semi-continuous denuder breakthrough determination.**

[Figure]

**Figure S2: Example of evaluation of denuder breakthrough contribution to the TC measurement uncertainty with inline Teflon filter method (see Fig. S1 (b) for TCA08 setup). Denuder breakthrough (TC$_{DB}$) is measured in chamber 1 where sample air stream passes denuder and inline Teflon filter. The TCA08 was operated on a 1-hour time-base, sampling PM2.5 fraction at 16.7 LPM. The measurement campaign was conducted between 18 December 2019 and 4 January 2020 at the urban background air quality monitoring of Aerosol d.o.o. company at 46.0715°N, 14.5018°E, elevation 302 m.**

---

## Author Response (AR3)

Dear Associate editor,

5 Thank you for your comments. Please find new version of the manuscript and supplement with your comments addressed below.

Martin Rigler

[revised manuscript text omitted]

**1. Example of in-house intercomparison results of TCA08 and AE33 instruments after their assembly to the reference set of instruments**

**Table S1. Example of intercomparison results of TCA08 and AE33 after their assembly to the reference set of instruments as one of the tests during final inspection procedure. In-house defined requirements for successful intercomparison between new and reference set of instruments are: 1. TCA08: TC concentration range up to 75.000 ng/m³, slope between 0.95-1.05, $R^2$ above 0.98 ; 2. AE33: eBC concentrations up to 25.000 ng/m³; slope between 0.95-1.05, $R^2$ above 0.98.**

| | TCA08 serial number | Slope | $R^2$ | N | $TC_{max}$ (ng/m³) | $TC_{min}$ (ng/m³) |
|---|---|---|---|---|---|---|
| 1 | TCA08-S00-0131 | 1.02 | 1.00 | 65 | 45856 | 2732 |
| 2 | TCA08-S00-0132 | 1.01 | 1.00 | 65 | 45803 | 2732 |
| 3 | TCA08-S00-0133 | 1.00 | 1.00 | 19 | 67274 | 6286 |
| 4 | TCA08-S00-0134 | 1.00 | 1.00 | 19 | 67274 | 6286 |
| 5 | TCA08-S00-0135 | 1.00 | 1.00 | 19 | 67274 | 6286 |
| 6 | TCA08-S00-0136 | 0.99 | 1.00 | 139 | 76684 | 2498 |
| 7 | TCA08-S00-0137 | 1.00 | 1.00 | 65 | 51119 | 2592 |
| 8 | TCA08-S00-0138 | 0.99 | 1.00 | 65 | 51119 | 2592 |
| 9 | TCA08-S00-0139 | 0.96 | 1.00 | 65 | 51119 | 2592 |
| 10 | TCA08-S00-0140 | 1.02 | 1.00 | 62 | 73190 | 2644 |
| 11 | TCA08-S00-0141 | 1.00 | 1.00 | 22 | 64579 | 3336 |
| 12 | TCA08-S00-0142 | 1.01 | 1.00 | 136 | 76466 | 2823 |

| | | Slope | R² | N | eBC_max (ng/m³) | eBC_min (ng/m³) |
|---|---|---|---|---|---|---|
| 13 | TCA08-S00-0143 | 1.00 | 1.00 | 22 | 64579 | 3336 |
| 14 | TCA08-S00-0144 | 1.00 | 1.00 | 22 | 64609 | 3336 |
| 15 | TCA08-S00-0145 | 0.97 | 1.00 | 137 | 72992 | 2823 |

| | AE33 serial number | Slope | R² | N | eBC_max (ng/m³) | eBC_min (ng/m³) |
|---|---|---|---|---|---|---|
| 1 | AE33-S08-01036 | 1.01 | 1.00 | 6368 | 19666 | 265 |
| 2 | AE33-S08-01037 | 1.00 | 1.00 | 6368 | 19478 | 220 |
| 3 | AE33-S08-01038 | 1.02 | 1.00 | 6368 | 21988 | 139 |
| 4 | AE33-S08-01039 | 1.02 | 1.00 | 6368 | 20126 | 198 |
| 5 | AE33-S08-01040 | 1.00 | 1.00 | 6368 | 21338 | 224 |
| 6 | AE33-S08-01041 | 0.99 | 1.00 | 6368 | 20384 | 231 |
| 7 | AE33-S08-01042 | 1.04 | 1.00 | 6320 | 22141 | 123 |
| 8 | AE33-S08-01043 | 1.04 | 1.00 | 6368 | 21380 | 286 |
| 9 | AE33-S08-01044 | 1.03 | 1.00 | 6368 | 21249 | 223 |
| 10 | AE33-S08-01045 | 1.02 | 1.00 | 6368 | 21050 | 172 |
| 11 | AE33-S08-01046 | 0.99 | 1.00 | 6368 | 20433 | 267 |
| 12 | AE33-S08-01047 | 1.03 | 1.00 | 6368 | 20583 | 230 |
| 13 | AE33-S08-01048 | 1.03 | 1.00 | 6368 | 21061 | 237 |
| 14 | AE33-S08-01049 | 1.03 | 1.00 | 6320 | 20699 | 249 |
| 15 | AE33-S08-01050 | 1.02 | 1.00 | 6320 | 20064 | 243 |

**2. TCA08 setup for semi-continuous denuder breakthrough determination**

[Figure]

**Figure S1. TCA08 setup when (a) sampling and (b) performing semi-continuous denuder breakthrough measurement. Note that the tubing length is identical in both setups. This permits the test to be performed at a permanent installation without disturbing the inlet plumbing.**

**3. Evaluation of denuder breakthrough contribution to the TC measurement uncertainty with inline Teflon filter method**

[Figure]

**Figure S2. Example of evaluation of denuder breakthrough contribution to the TC measurement uncertainty with inline Teflon filter method (see Fig. S1 (b) for TCA08 setup). Denuder breakthrough (TC$_{DB}$) is measured in chamber 1 where sample air stream passes denuder and inline Teflon filter. The TCA08 was operated on a 1-hour time-base, sampling PM$_{2.5}$ fraction at 16.7 LPM. The measurement campaign was conducted between 18 December 2019 and 4 January 2020 at the urban background air quality monitoring of Aerosol d.o.o. company at 46.0715°N, 14.5018°E, elevation 302 m.**

---

## Author Response (AR4)

Dear Associate editor.

Thank you for your comments. Please find new version of the manuscript and supplement with your comments addressed below.

Martin Rigler

[revised manuscript text omitted]

---

## Author Response (AR5)

Dear Associate editor.

Thank you for accepting our manuscript. Please find manuscript with the small correction bellow:
- Page 22, line 14. "camping" was replaced by "campaign".

Martin Rigler

[revised manuscript text omitted]